# Dimer-monomer transition defines a hyper-thermostable peptidoglycan hydrolase mined from bacterial proteome by lysin-derived antimicrobial peptide-primed screening

Li Zhang[1,2†], Fen Hu[3†], Zirong Zhao[1†], Xinfeng Li[4], Mingyue Zhong[4], Jiajun He[1], Fangfang Yao[5], Xiaomei Zhang[1], Yuxuan Mao[1], Hongping Wei[4], Jin He[1]*, Hang Yang[4,6,7]*

[1]National Key Laboratory of Agricultural Microbiology & Hubei Hongshan Laboratory, College of Life Science and Technology, Huazhong Agricultural University, Wuhan, China; [2]College of Veterinary Medicine, Henan University of Animal Husbandry and Economy, Zhengzhou, China; [3]Key Laboratory of Gastrointestinal Cancer (Fujian Medical University), Ministry of Education, Department of Etiology, School of Basic Medical Sciences, Fujian Medical University, Fuzhou, China; [4]Key Laboratory of Virology and Biosafety, Wuhan Institute of Virology, Chinese Academy of Sciences, Wuhan, China; [5]The State Key Laboratory Breeding Base of Basic Science of Stomatology (Hubei-MOST) & Key Laboratory of Oral Biomedicine, Ministry of Education, School of Stomatology, Wuhan University, Wuhan, China; [6]University of Chinese Academy of Sciences, Beijing, China; [7]Hubei Jiangxia Laboratory, Wuhan, China

*For correspondence:
hejin@mail.hzau.edu.cn (JH);
yangh@wh.iov.cn (HY)

†These authors contributed equally to this work

## eLife assessment

This **valuable** study explores a new strategy of lysin-derived antimicrobial peptide-primed screening to find peptidoglycan hydrolases from bacterial proteomes. Using this strategy, the authors identified five peptidoglycan hydrolases from Acinetobacter baumannii, which they tested on various Gram-positive and Gram-negative pathogens for antimicrobial activity. The revised manuscript addressed most of the prior concerns, and the data presented are **solid** and will be of interest to microbiologists.

**Abstract** Phage-derived peptidoglycan hydrolases (i.e. lysins) are considered promising alternatives to conventional antibiotics due to their direct peptidoglycan degradation activity and low risk of resistance development. The discovery of these enzymes is often hampered by the limited availability of phage genomes. Herein, we report a new strategy to mine active peptidoglycan hydrolases from bacterial proteomes by lysin-derived antimicrobial peptide-primed screening. As a proof-of-concept, five peptidoglycan hydrolases from the Acinetobacter baumannii proteome (PHAb7-PHAb11) were identified using PlyF307 lysin-derived peptide as a template. Among them, PHAb10 and PHAb11 showed potent bactericidal activity against multiple pathogens even after treatment at 100°C for 1 hr, while the other three were thermosensitive. We solved the crystal structures of PHAb8, PHAb10, and PHAb11 and unveiled that hyper-thermostable PHAb10 underwent a unique

folding-refolding thermodynamic scheme mediated by a dimer-monomer transition, while thermo-sensitive PHAb8 formed a monomer. Two mouse models of bacterial infection further demonstrated the safety and efficacy of PHAb10. In conclusion, our antimicrobial peptide-primed strategy provides new clues for the discovery of promising antimicrobial drugs.

## Introduction

Despite decades of concerted action from all over the world, antimicrobial resistance remains a growing global threat today (*Nathan, 2020*). The misuse and overuse of antimicrobials leads to their accumulation in the environment, eventually increasing the chances of bacterial pathogens to acquire resistance (*Van Boeckel et al., 2019*; *Rawson et al., 2020*). Therefore, the concept of 'One Health' has been proposed to address the alarming levels of antimicrobial resistance (*McEwen and Collignon, 2018*; *Hernando-Amado et al., 2019*). In recent decades, the number of bacteria resistant to first-line drugs, as well as second- and third-line drugs, has been increasing. Some of these bacteria have even been identified as multidrug-resistant, extensively drug-resistant, or pan-drug-resistant isolates (*Talaat et al., 2022*). For example, methicillin-resistant *Staphylococcus aureus*, vancomycin-resistant *S. aureus*, vancomycin-resistant *Enterococcus*, colistin-resistant Enterobacteriaceae, third-generation cephalosporin-resistant *Escherichia coli* and *Klebsiella pneumoniae*, carbapenem-resistant *E. coli*, *K. pneumoniae*, and *Pseudomonas aeruginosa*, and multidrug-resistant *Acinetobacter baumannii* pose a serious threat and are the leading cause of morbidity and mortality in human infectious diseases (*Brink, 2019*). In addition, the global medical need for new antimicrobials has not yet been fully addressed.

Considerable evidence shows that bacterial peptidoglycan is a promising target for the development of antimicrobial agents with a low risk of resistance development. A well-documented example in this regard are the peptidoglycan hydrolases, also known as lysins, derived from phages. In recent years, lysins have been shown to be efficient alternatives to traditional antibiotics for infections caused by Gram-positive bacteria in various animal models (*De Maesschalck et al., 2020*; *Schmelcher and Loessner, 2021*). Encouragingly, several lysins targeting *S. aureus* have been evaluated in clinical trials (*Theuretzbacher and Piddock, 2019*). However, the development of lysins targeting Gram-negative bacteria is relatively recent, mainly because the bacterial outer membrane prevents lysins from accessing their peptidoglycan substrates. In the last decade, several strategies have been developed to enable lysin to overcome the outer membrane barrier and directly target Gram-negative bacteria. For instance, Artilysins are constructed by fusion of cationic nonapeptides to effectively eradicate resistant and persistent *A. baumannii* (*Defraine et al., 2016*); Innolysins are engineered by fusion of receptor-binding proteins to display bactericidal activity against *E. coli* that was resistant to third-generation cephalosporins (*Zampara et al., 2020*); bioengineered lysin-bacteriocin fusion molecule, i.e., Lysocins, are able to deliver lysin across the outer membrane of Gram-negative bacteria and show selective anti-Pseudomonal activity (*Heselpoth et al., 2019*). In addition, the establishment of high-throughput mining or engineering strategies has accelerated the discovery of active lysins against Gram-negative pathogens, e.g., the VersaTile-driven platform, which can rapidly screen engineered lysins active against *A. baumannii* from tens of thousands of combinations by integrating new DNA assembly methods and iterative screening procedures (*Gerstmans et al., 2020*). However, compared with the current progress in the clinical translation of lysins against Gram-positive bacteria, the discovery of lysins against Gram-negative bacteria that meet the needs described in the WHO priority pathogen list is still urgently needed (*Briers and Lavigne, 2015*; *Lai et al., 2020*).

Additionally, the discovery of phage-derived peptidoglycan hydrolases has also been hampered by limited sources of published phage genome data. Although not yet tested extensively, recent progress in lysins targeting Gram-negative bacteria have demonstrated a clue linking antibacterial activity to their internal antimicrobial peptides (*Vázquez et al., 2021*), by which lysins cross the bacterial outer membrane and contact its underlying peptidoglycan substrates (*Lood et al., 2015*; *Thandar et al., 2016*; *Li et al., 2021*). These observations prompted us to consider whether antibacterial peptidoglycan hydrolases or their homologs could be mined from bacterial proteomes using lysin-derived antimicrobial peptide as template. As a proof-of-concept, we here used the antimicrobial peptide P307 derived from PlyF307 lysin (*Lood et al., 2015*) as a template to search for peptidoglycan hydrolases from the *A. baumannii* proteome database (PHAbs, *Figure 1—figure supplement*

**eLife digest** Bacteria are increasingly becoming resistant to antibiotics, leading to a rise in cases of dangerous diseases around the world. Innovative treatments are urgently needed, prompting researchers to turn to alternative approaches.

One strategy is to focus on finding new ways to target the bacterial cell wall, a shield-like structure that wraps around the cell and protects it against the environment. Certain hydrolase enzymes can weaken this wall by breaking down its primary component, a type of molecules known as peptidoglycans. Lysins, for example, are produced by phage viruses (which prey on bacteria) and have gathered momentum as antimicrobial agents.

However, clinical studies so far have largely focused on using peptidoglycan hydrolases against Gram-positive bacteria, in which peptidoglycans are directly exposed to the environment. In Gram-negative species, on the other hand, the cell wall includes an outer membrane that makes it harder for the enzymes to access their targets. Finding better phage-derived lysins that work against Gram-negative bacteria has been challenging so far, partly because this would require scanning the genomes of various phage species for candidates – an information that is difficult to access.

Instead, Zhang et al. turned to peptidoglycan hydrolases produced by bacteria themselves, for example to eliminate competitors. To find these enzymes, the team used a specific structure in a phage-derived lysin as a template; known as P307, this short sequence (or peptide) allows the viral enzyme to cross the outer membrane of Gram-negative bacteria and reach the peptidoglycans below.

Using a range of computational method, P307 was screened against the pool of proteins produced by the bacterium *Acinetobacter baumannii*. This revealed five bacterial peptidoglycan hydrolases, two of which had the potent ability to kill harmful species of Gram-positive and Gram-negative bacteria both in vitro and in mice. They retained this ability even after having been exposed to high temperatures, with further experiments pointing to unique structural properties underlying this stability. Taken together, these findings highlight an effective method to identify new bacterial-derived peptidoglycan hydrolases that could serve as antimicrobial agents.

*1*). Two of them (PHAb10 and PHAb11) were found to be hyperthermally stable even after treatment at 100°C for 1 hr, which is mediated by a unique monomer-dimer swapping. Furthermore, both were active against a wide variety of clinically relevant bacteria in vitro and in mouse infection models. This observation supports our hypothesis that antimicrobial peptide-primed mining strategy is feasible for discovering new peptidoglycan hydrolases with high bactericidal activity and desirable physicochemical properties from bacterial proteomes.

## Results

### Screening of putative peptidoglycan hydrolases from the *A. baumannii* proteome based on a lysin-derived antimicrobial peptide-primed mining strategy

Since a variety of lysins active against Gram-negative bacteria are shown to harbor an N- or C-terminal antimicrobial peptide, this led us to speculate that antimicrobial peptide could be used as template to mine new peptidoglycan hydrolases. To prove this hypothesis, P307, a well-documented antimicrobial peptide against *A. baumannii* from PlyF307 lysin (***Lood et al., 2015***; ***Thandar et al., 2016***), was used as the template to mine putative peptidoglycan hydrolases from *A. baumannii* (PHAb) proteome databases publicly available in NCBI (***Figure 1a***). Finally, 204 hits were identified and further divided into five clades based on phylogenetic analysis. Clade I contained 162 hits with high similarity (>85%) to the AcLys lysin from *A. baumannii* 5057UW prophage (***Figure 1b***), and clade II–V contained 42 hits with relatively lower homology (40–77%) to each other (***Figure 1b***; ***Figure 1—figure supplement 2a***).

Due to the high similarity of amino acid sequences within each clade, we selected a putative peptidoglycan hydrolase from each clade for further characterization by analyzing the selectivity priority in evolution and the overall physicochemical properties (water solubility, hydrophilicity, hydrophobicity, charge, flexibility, and rigidity) of each sequence in silico. Finally, WP_065188953.1 from clade I, WP_038349544.1 from clade II, KCY28522.1 from clade III, WP_000109885.1 from clade IV, and

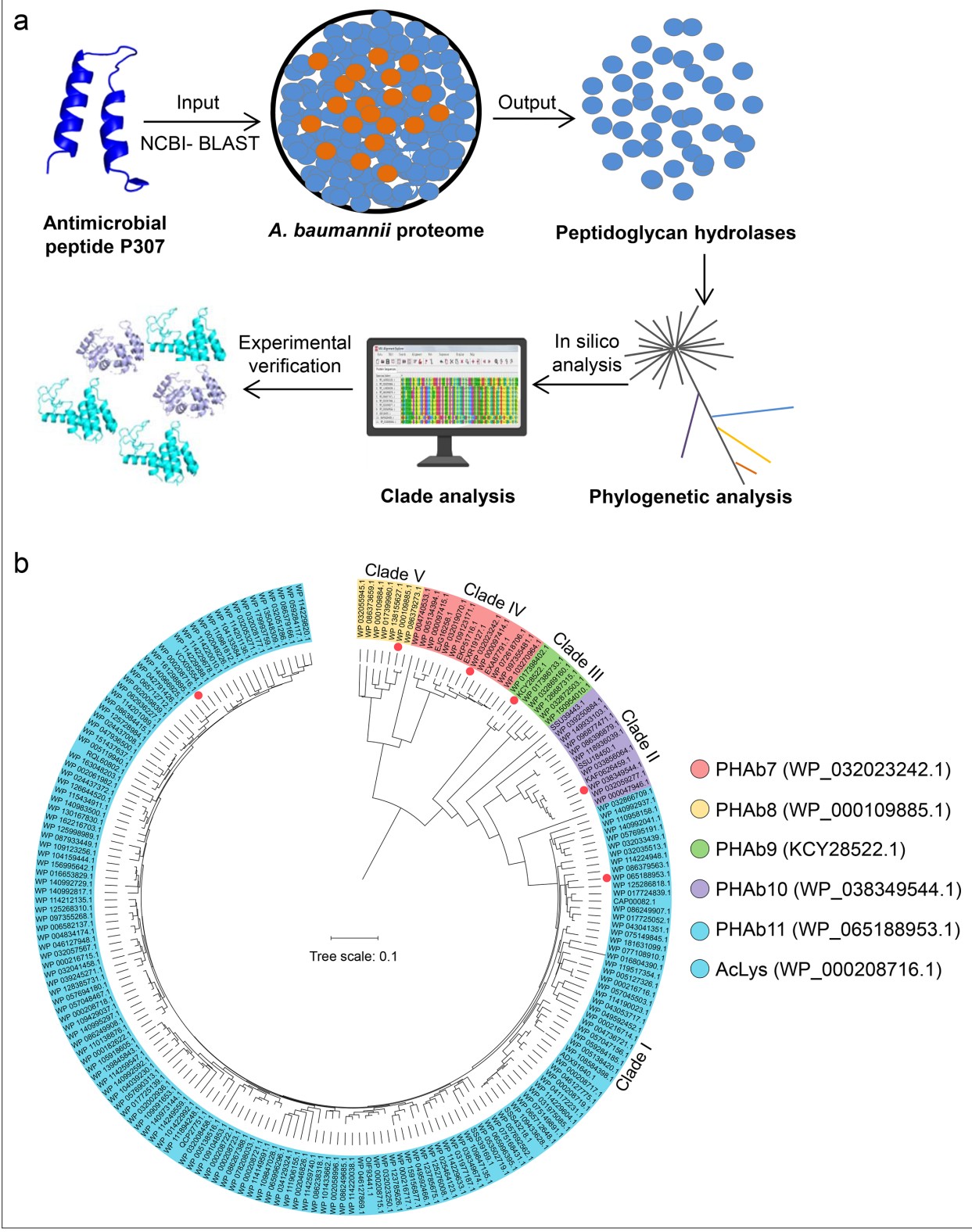

**Figure 1.** Screening of putative peptidoglycan hydrolases from the *A. baumannii* proteome. (**a**) Workflow for the development of an integrated system for peptidoglycan hydrolase screening. (**b**) Phylogenetic analysis of putative peptidoglycan hydrolases from the *A. baumannii* proteome database. Multiple sequence alignments are performed using MEGAX software with ClustalW algorithm, and the neighbor-joining method is used to construct a phylogenetic tree. Classified clades are labeled in different colors and representative peptidoglycan hydrolases from each clade are pointed by colored circles.

*Figure 1 continued on next page*

*Figure 1 continued*

The online version of this article includes the following figure supplement(s) for figure 1:

**Figure supplement 1.** Workflow of the search integrated system for peptidoglycan hydrolases.

**Figure supplement 2.** Selective five putative peptidoglycan hydrolases.

WP_032023242.1 from clade V were shortlisted and renamed as PHAb11, PHAb10, PHAb9, PHAb8, and PHAb7, respectively, for further investigation (*Figure 1b*). Further homology analysis revealed that these five peptidoglycan hydrolases showed great diversity with the structured AcLys lysin and the original PlyF307 lysin (*Figure 1—figure supplement 2b*).

## PHAb10 and PHAb11 are highly thermostable peptidoglycan hydrolases with a broad spectrum of action

To test the activity of these PHAbs, their coding sequences were chemically synthesized, cloned into a vector, and expressed in *E. coli* cells. As expected, all five PHAbs were well expressed as soluble proteins in *E. coli* BL21(DE3) with high purity (*Figure 2—figure supplement 1a*). Next, we examined the hydrolytic activity of these five PHAbs on peptidoglycan derived from *A. baumannii* 3437 by a halo assay (*Vander Elst et al., 2020*). The results showed that all PHAbs form clear zones on the autoclaved *A. baumannii* lawns even at a low concentration of 2.5 μg/ml (*Figure 2—figure supplement 1b*), indicating that these PHAbs are active against peptidoglycan of *A. baumannii*. To further confirm the ability of these PHAbs against peptidoglycan from different bacteria, we measured the area of clear zones generated by different concentrations of PHAbs on each lawn of two isolates of *A. baumannii*, *P. aeruginosa*, and *E. coli*. Encouragingly, these PHAbs were active against peptidoglycans from all strains tested and formed a clear halo (*Figure 2a*). Among these PHAbs, PHAb10 and PHAb11 showed larger size of clear zones under each concentration tested (*Figure 2a*), indicating that these two PHAbs might have stronger peptidoglycan hydrolyzing activity. To prove this, we further evaluated the bactericidal activity of these PHAbs against each isolate of *A. baumannii*, *P. aeruginosa*, and *E. coli* by a log-killing assay. Consistent with the halo assay, PHAb10 and PHAb11 exhibited outstanding bactericidal activity against all strains tested, with a 3.5–4.6 log reduction in viable bacterial counts after treatment for 1 hr with 50 μg/ml of each enzyme (*Figure 2b*). Like other lysins, the potent bactericidal activity of PHAb10 and PHAb11 was time-dependent. After treatment with 50 μg/ml of either enzyme, a ~0.9 log reduction in viable *A. baumannii* 3437 for both enzymes were observed within the first 30 s, with a substantial reduction to 3.1 logs (from 6.0 log to 2.9 log) for PHAb10 and 2.5 logs (from 6.0 log to 3.5 log) for PHAb11 after 15 min (*Figure 2c*).

Since previous reports documented the diverse performance of lysins against Gram-negative and Gram-positive bacterial species in exponential and stationary phases (*Lood et al., 2015*), we therefore examined the bactericidal activity of PHAb10 and PHAb11 against different bacteria in different growth phases. Interestingly, robust bactericidal activity was observed for both enzymes in exponential and stationary phase cultures of *A. baumannii*, *P. aeruginosa*, *E. coli*, *K. pneumoniae*, *Streptococcus suis*, and *E. faecalis* (*Figure 2d*). It should also be noted that exponential bacteria were more susceptible to these two enzymes than stationary bacteria, similar to observations reported elsewhere (*Lood et al., 2015*). However, for *S. aureus*, detectable susceptibility to PHAb11 was only observed in its exponential phase (*Figure 2d*). In addition, PHAb10 and PHAb11 exhibited dose-dependent killing against sensitive Gram-negative and Gram-positive strains tested. Specifically, treatment with 6.25 μg/ml PHAb10 for 1 hr resulted in a ~3 log reduction in *A. baumannii*, a ~2.7 log reduction in *P. aeruginosa*, a ~3 log reduction in *E. coli*, a ~1.08 log reduction in *K. pneumoniae*, and a ~0.94 log reduction in *S. suis*, and treatment with 6.25 μg/ml PHAb11 for 1 hr caused a ~2.8 log reduction in *A. baumannii*, a ~2.6 log reduction in *P. aeruginosa*, a ~2.7 log reduction in *E. coli*, a ~0.69 log reduction in *K. pneumoniae*, a ~1.3 log reduction in *S. aureus*, and a ~2.15 log reduction in *S. suis* (*Figure 2e*). Interestingly, *E. faecalis* was more sensitive to PHAb10 and PHAb11, showing a ~3.8 log and a ~4.6 log reduction respectively after treatment with 10 μg/ml of each enzyme for 1 hr (*Figure 2f*).

To further confirm the bactericidal activity of PHAb10 and PHAb11, the susceptibility of various clinical isolates such as *A. baumannii*, *P. aeruginosa*, *E. coli*, *K. pneumonia*, *S. aureus*, and *E. faecalis*, as well as representative isolates of *Streptococcus dysgalactiae*, *Streptococcus agalactiae*, *Streptococcus pyogenes*, *Streptococcus pneumoniae*, and *S. suis* to both enzymes were examined by

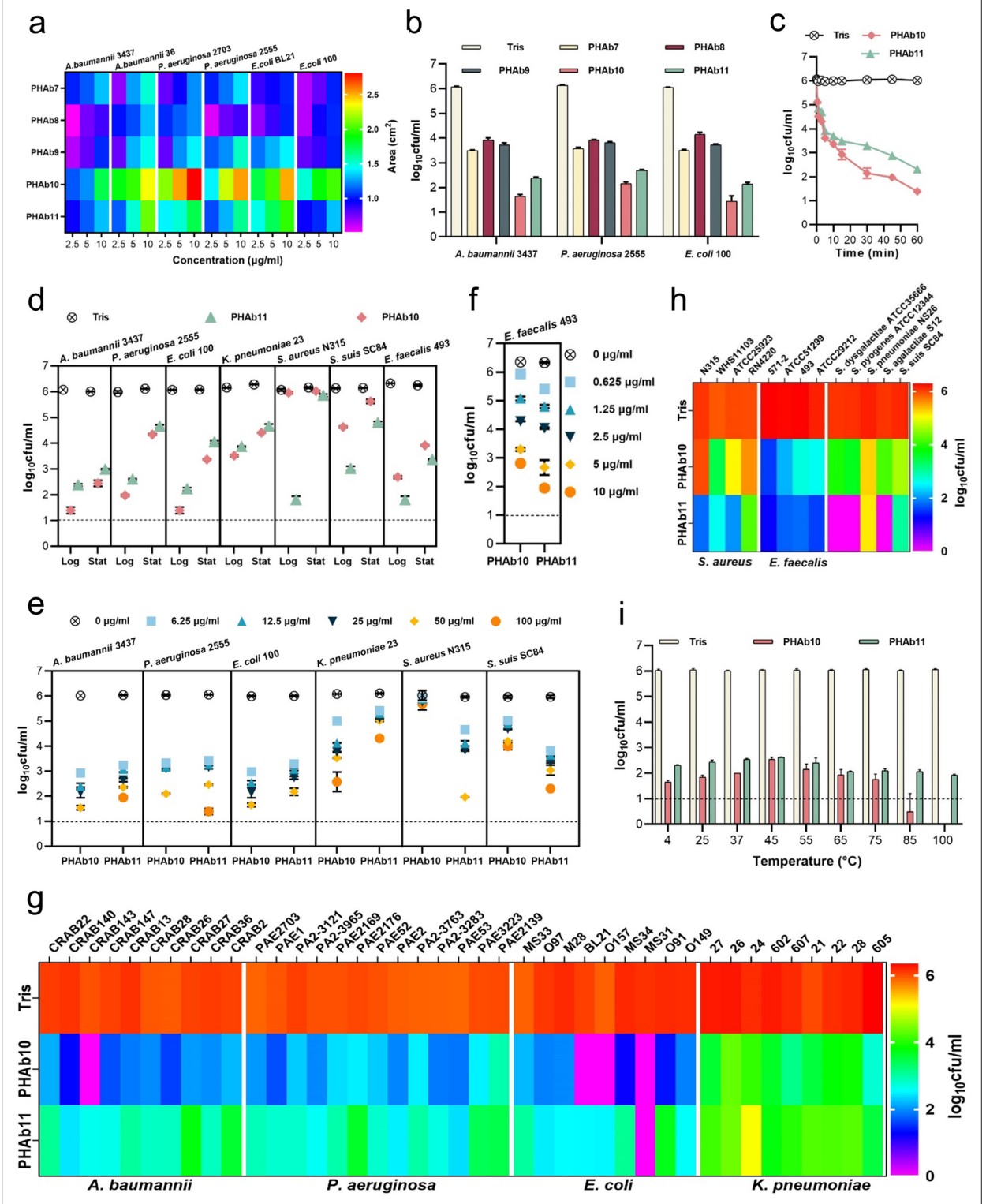

**Figure 2.** PHAb10 and PHAb11 are highly thermostable peptidoglycan hydrolases with a broad spectrum of action. (**a**) Heat map showing the halo size formed by different concentrations of peptidoglycan hydrolases on different bacterial lawns. (**b**) Bactericidal activity of five peptidoglycan hydrolases (50 μg/ml) in 20 mM Tris-HCl (pH 7.4) at 37°C for 1 hr against different exponential bacteria. (**c**) Time-dependent bactericidal activity of 50 μg/ml PHAb10 or PHAb11 against exponential *A. baumannii* in 20 mM Tris-HCl (pH 7.4). (**d**) Bactericidal activity of 50 μg/ml PHAb10 or PHAb11 in 20 mM Tris-HCl (pH 7.4) at 37°C for 1 hr against different bacteria in exponential and stationary phases. (**e–f**) Dose-dependent bactericidal activity of PHAb10 and PHAb11 against multiple exponential bacteria in 20 mM Tris-HCl (pH 7.4) at 37°C for 1 hr. (**g–h**) Susceptibility of various Gram-negative (**g**) and Gram-positive

*Figure 2 continued on next page*

*Figure 2 continued*

(**h**) bacterial strains to PHAb10 and PHAb11. Exponential cultures of each bacterium are treated with 50 µg/ml PHAb10 or PHAb11 for 1 hr at 37°C and residual viable bacterial cells are counted by plating serial dilutions onto agar plates. For *Enterococcus faecalis*, 10 µg/ml of each peptidoglycan hydrolase is used. (**i**) Thermal stability of PHAb10 and PHAb11. Each enzyme is stored at different temperatures for 1 hr, cooled to room temperature, and then incubated with exponential *A. baumannii* 3437 in 20 mM Tris-HCl (pH 7.4) at a final concentration of 50 µg/ml at 37°C for 1 hr. Viable bacteria are counted after each treatment by plating serial dilutions on Lysogeny Broth (LB) agar. All assays were performed with at least three biological replicates (n=3-9). Dash lines represent the limit of detection and data below the limit of detection is not shown.

The online version of this article includes the following source data and figure supplement(s) for figure 2:

**Figure supplement 1.** Testing of the bactericidal activity of the screened putative peptidoglycan hydrolases.

**Figure supplement 1—source data 1.** PDF file containing original SDS-PAGE gel for *Figure 2—figure supplement 1*, indicating the relevant bands.

**Figure supplement 1—source data 2.** Original file for SDS-PAGE analysis displayed in *Figure 2—figure supplement 1*.

log-killing assay. Our results showed that all tested strains, whether antibiotic-resistant or not, were sensitive to PHAb10 and PHAb11 (*Figure 2g and h*). Importantly, Gram-negative bacteria were more sensitive to PHAb10, with 3.9–6.0 log reductions for *A. baumannii* isolates, 3.7–5.0 log reductions for *E. coli* isolates, 2.4–4.3 log reductions for *P. aeruginosa* isolates, and 1.1–3.6 log reductions for *K. pneumoniae* isolates (*Figure 2g*). In contrast, PHAb11 was more bactericidal than PHAb10 against Gram-positive bacteria, causing 1.8–4.3 log reductions for *S. aureus* isolates, 4.5–5.0 log reductions for *E. faecalis* isolates, and 1.0–6.1 log reductions for various streptococcal strains (*Figure 2h*).

Further biochemical characterization showed that PHAb10 and PHAb11 remained highly active at pH 5.0–10.0, and the maximum bactericidal activity was observed at pH 7.0 for both enzymes (*Figure 2—figure supplement 1c*). Similar to other previously reported lysins (*Larpin et al., 2018*; *Kim et al., 2020*), PHAb10 and PHAb11 were sensitive to NaCl and urea in a dose-dependent inhibitory manner (*Figure 2—figure supplement 1d*). Surprisingly, both enzymes retained outstanding bactericidal activity after heat treatment at temperature up to 100°C for 1 hr (*Figure 2i*). Indeed, after 1 hr treatment at 100°C, the bactericidal activity was improved slightly for both enzymes. In contrast, PHAb7, PHAb8, and PHAb9 completely lost their activity after treatment at 70°C for 1 hr (*Figure 2—figure supplement 1e*). Together, these observations showed that PHAb10 and PHAb11 were highly thermostable peptidoglycan hydrolases with a broad spectrum of action.

## Different action mechanisms of PHAb10 and PHAb11 in killing Gram-negative and Gram-positive bacteria

Due to the natural barrier of the outer membrane, several peptidoglycan hydrolases (i.e. lysins) have been reported to have no or low bactericidal activity against Gram-negative bacteria in the absence of outer membrane penetrating agents (*Briers and Lavigne, 2015*; *Ghose and Euler, 2020*; *Lai et al., 2020*). However, for most of these lysins, their activity are almost exclusively restricted to Gram-negative bacteria (*Guo et al., 2017*; *Khan et al., 2021*), raising questions about the mechanisms of action of PHAb10 and PHAb11 in killing both Gram-negative and Gram-positive bacteria. Structural prediction analysis showed that PHAb10 contains a putative lysozyme catalytic domain (amino acids 1–110, P10-Lys) and a C-terminal cationic peptide (amino acids 111–149, P10-CP), whereas PHAb11 contains an N-terminal cationic peptide with unknown function (amino acids 1–36, P11-NP), followed by a putative lysozyme catalytic domain (amino acids 37–145, P11-Lys, with 68.5% sequence similarity with P10-Lys) and a C-terminal cationic peptide (amino acids 146–184, P11-CP, which shows 94.9% sequence similarity with P10-CP) (*Figure 3a*). Studies have shown that the cationic region of phage lysins can facilitate their penetration into the outer membrane, thereby fulfilling their bactericidal activity against Gram-negative bacteria (*Lood et al., 2015*; *Larpin et al., 2018*; *Li et al., 2021*). To examine whether it is the reason for the killing ability of PHAb10 and PHAb11 against Gram-negative bacteria, we further analyzed in silico the physicochemical properties of these putative cationic peptides harbored by these two enzymes and found that all three peptides, i.e., P10-CP, P11-CP, and P11-NP might act as antimicrobial peptides (*Supplementary file 1a*). Therefore, we synthesized these peptides and examined their antibacterial activity by log-killing assay. As shown in *Figure 3b*, we observed bactericidal activity in P10-CP and P11-CP against *A. baumannii*, *P. aeruginosa*, *S. aureus*, and *E. faecalis* in both the exponential and stationary phases. However, rare lytic activity was observed in P11-NP, suggesting that its function remains to be established. Notably, P11-CP was more robust

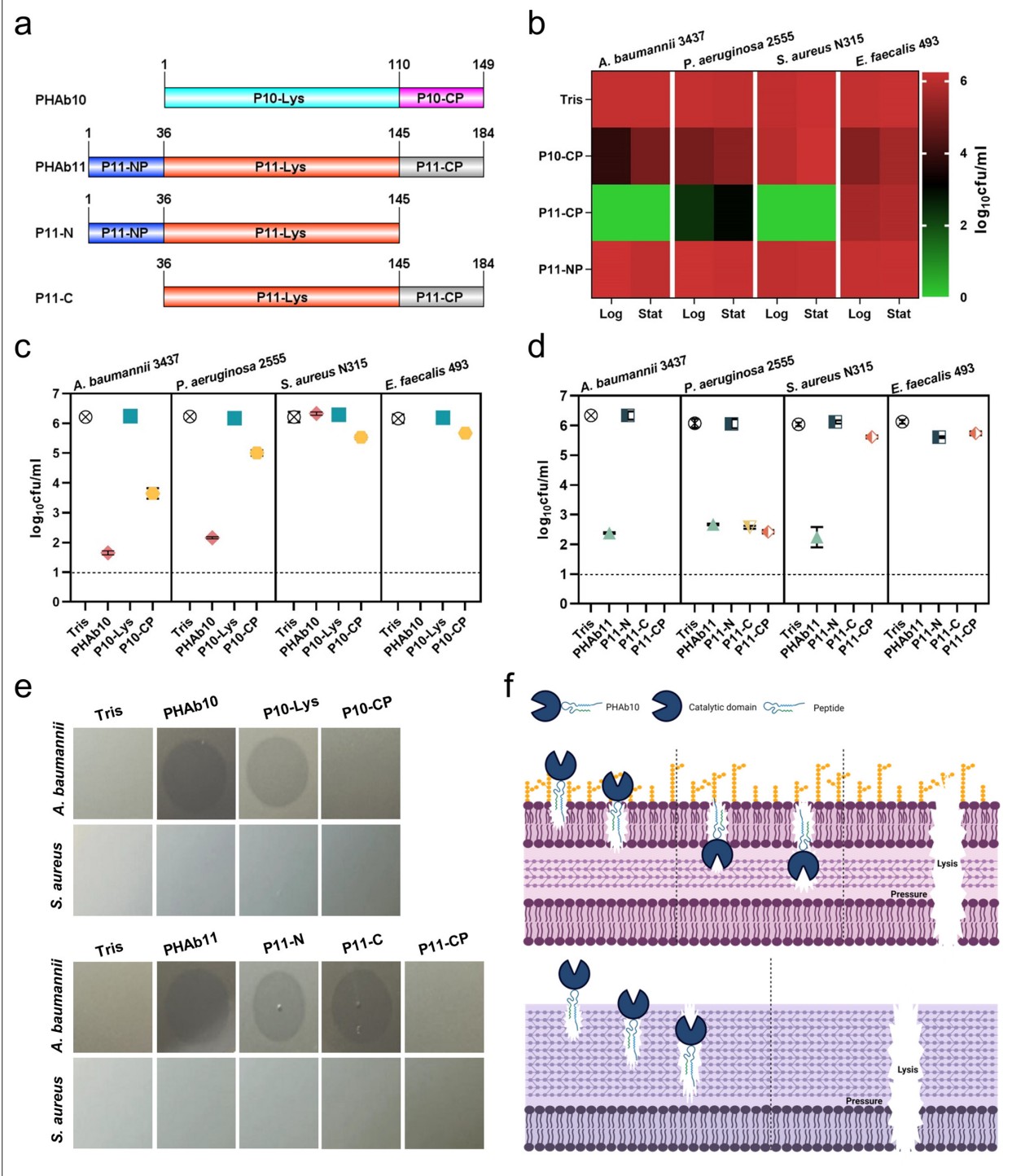

**Figure 3.** Mechanism of action of PHAb10 and PHAb11 on Gram-negative and Gram-positive bacteria. (**a**) Schematic representation of PHAb10 and PHAb11 domains. (**b**) Bactericidal activity of peptides derived from PHAb10 and PHAb11. Bacterial cells are treated with 50 µg/ml P10-CP, P11-NP, or P11-CP in 20 mM Tris-HCl at 37°C for 1 hr. (**c–d**) Bactericidal activity of different truncations of PHAb10 and PHAb11. Exponential *A. baumannii* 3437 cells are incubated with 50 µg/ml of each truncated fragment in 20 mM Tris-HCl for 1 hr at 37°C. Dash lines represent the limit of detection and data below the limit of detection is not shown. (**e**) Peptidoglycan hydrolytic activity of different truncations of PHAb10 and PHAb11. 0.1 µg of each truncated fragment is dropped onto autoclaved lawns of *A. baumannii* 3437 and *S. aureus* N315 and incubated at 37°C for 4 hr. The groups treated with an equal volume of 20 mM Tris-HCl served as controls. (**f**) Schematic diagram of the bactericidal mechanisms of PHAb10 against Gram-negative bacteria (top) and Gram-positive bacteria (bottom). All assays (**b-e**) were performed with at least three biological replicates (n=3-9).

than P10-CP (*Figure 3b*), which might be due to the presence of an additional positively charged amino acid (i.e. lysine) in P11-CP (*Supplementary file 1a*). More importantly, the putative lysozyme catalytic domains of two enzymes, P10-Lys, P11-Lys, and P11-N showed rare bactericidal activity against all Gram-negative and Gram-positive bacteria tested (*Figure 3c and d*), which indicated that the antimicrobial peptides, namely P10-CP and P11-CP, were essential for the bactericidal activity of both enzymes. Curiously, for unknown reasons, P11-C showed higher bactericidal activity than the holoenzyme PHAb11 against all bacteria tested (*Figure 3d*). Considering the different targets of antimicrobial peptides and peptidoglycan hydrolases, we further questioned the killing mechanism of PHAb10 and PHAb11 against susceptible bacteria. To this end, we evaluated the effects of different truncations of PHAb10 and PHAb11 on different bacteria by halo assay to test their peptidoglycan hydrolase activity. Results showed that lysozyme-containing truncations from both enzymes showed hydrolytic activity only toward peptidoglycan from Gram-negative bacteria but not from Gram-positive bacteria (*Figure 3e*), suggesting that the killing of Gram-positive bacteria by PHAb10 and PHAb11 is probably mediated mainly by their internal antimicrobial peptides, i.e., P10-CP and P11-CP. Supporting to this assumption, rare peptidoglycan hydrolytic activity was observed in P10-CP and P11-CP (*Figure 3e*). Altogether, these results suggested that the killing of Gram-negative bacteria by PHAb10 and PHAb11 might be a combination of lysozyme-mediated peptidoglycan hydrolytic activity and antimicrobial peptide-mediated outer membrane-disruption activity. However, their bactericidal activity against Gram-positive bacteria was primed solely by their internal antimicrobial peptides (*Figure 3f*).

## PHAb8 is a thermosensitive monomer, while PHAb10 and PHAb11 formed a thermostable dimer

To understand the catalytic mechanism and thermal stability of these peptidoglycan hydrolases, we attempted to decipher their structures and eventually obtained the crystal structures of PHAb8, PHAb10, and PHAb11 by X-ray crystallography (*Supplementary file 1b*). Results showed that PHAb8 was a monomer (*Figure 4a*; *Figure 4—figure supplement 1a*), while PHAb10 (*Figure 4b*) and PHAb11 (*Figure 4c*) formed an asymmetric dimer. Specifically, the two asymmetric units of PHAb10 formed an antiparallel dimer with a rotation angle of 177.2° along the rotation axis and a distance of 10.5 Å between them. Further, PHAb10 dimer adopted a circularly permuted architecture, in which the N-terminus of one chain and the C-terminus of another chain formed a cavity, which might be the binding site of the peptidoglycan substrate (*Figure 4b*). The two subunits of PHAb11 formed a tail-to-tail dimer with a rotation angle of −179.4° along the rotation axis and a distance of 37.1 Å between them. Each PHAb11 monomer contained an independent substrate-binding cavity around its catalytic triad (*Figure 4c*). Notably, the N-terminal 36 amino acids, P11-NP, were missing in our PHAb11 dimer. We speculated that it may be destroyed during crystal formation. Interestingly, PHAb8 showed high topological similarity to monomeric PHAb10 and PHAb11, with the typical catalytic triad (Glu-Asp-Thr) located at the coil region (*Figure 4d*; *Figure 4—figure supplement 1a–c*). Dali-based screening further showed that the catalytic triad of PHAb8, PHAb10, and PHAb11 were highly conserved in homologous proteins, including AcLys (PDB ID 6ET6), LysF1 (PDB ID 7M5I), SpmX-Mur-*Ae* (PDB ID 6H9D), P22 (PDB ID 2ANX), and R21 (PDB ID 3HDE) (*Figure 4—figure supplement 1d*; *Supplementary file 1c*). Superimposition of PHAb8, PHAb10, PHAb11 with other well-known lysozymes, including hen egg white lysozyme (HEWL, PDB ID 4HPI), goose egg white lysozyme (GEWL, PDB ID 153L), T4 lysozyme (T4L, PDB ID 1LYD), and lambda lytic transglycosylase (PDB ID 1LYD) revealed that the overall structures of PHAb8, PHAb10, and PHAb11 had relatively low similarity to HEWL, GEWL, and lambda lytic transglycosylase (*Figure 4—figure supplement 1e–g*). Despite their lower amino acid sequence similarity (*Figure 4—figure supplement 1h*), PHAb8, PHAb10, and PHAb11 superimposed well with T4 lysozyme, especially in their Glu-Asp-Thr catalytic triad regions (*Figure 4—figure supplement 1i*; *Supplementary file 1c*), suggesting that PHAb8, PHAb10, and PHAb11 may be phage-originated. Supporting to this, all three enzymes were predicted to be originated from prophages by PhageBoost (*Figure 4—figure supplement 2*). All these observations together indicated that PHAb8, PHAb10, and PHAb11 were highly evolutionarily related to T4 lysozyme and might share similar catalytic mechanisms of action.

To reveal which domain contributes to the thermal stability of PHAb10 and PHAb11, we examined the residual bactericidal activity of their individual domains by treating them at 100°C for 1 hr. Results

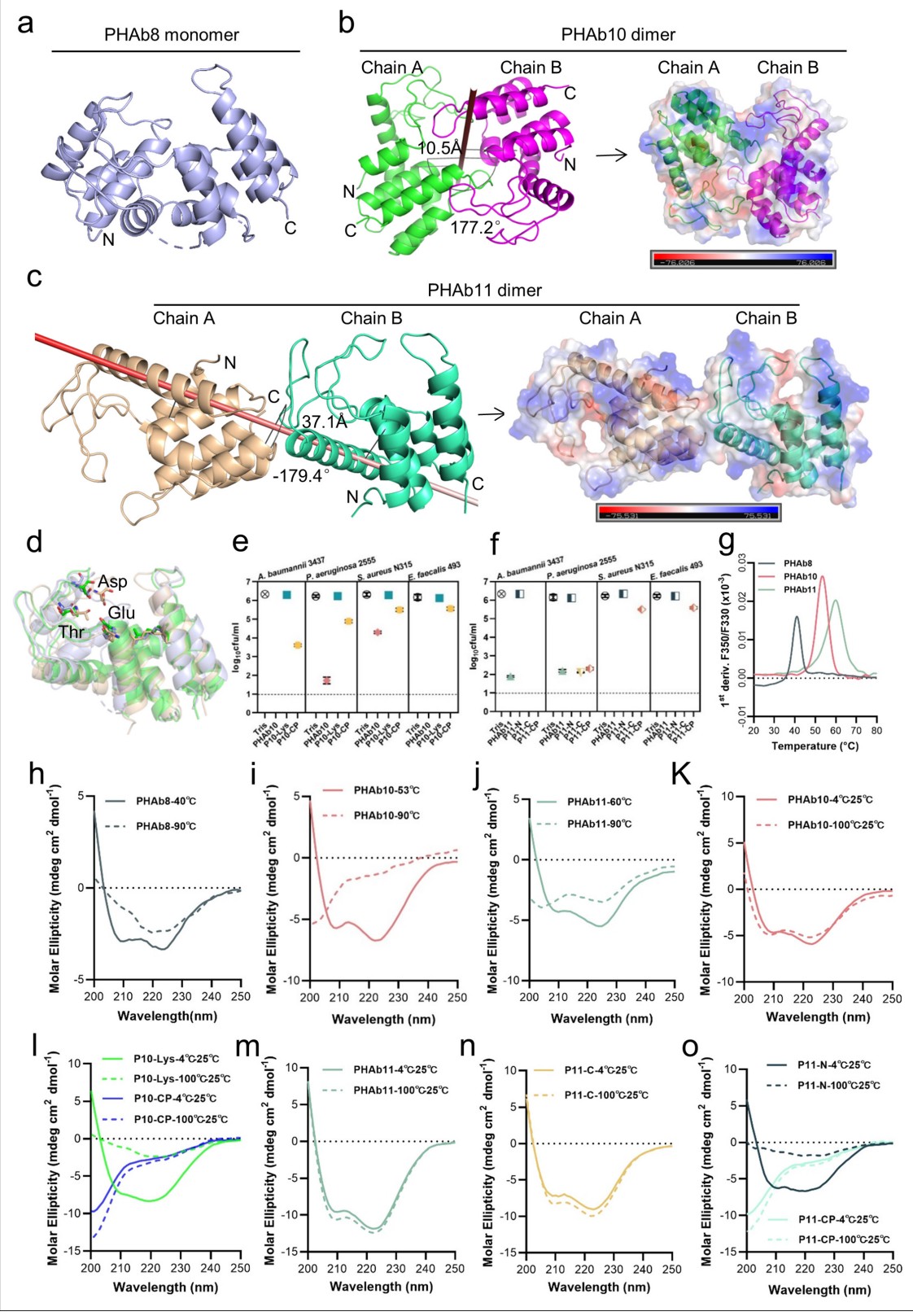

**Figure 4.** PHAb10 and PHAb11 are thermostable dimers. (**a**) Overall structure of PHAb8 monomer. (**b**) Dimeric structure of PHAb10. Chain A is colored in green, Chain B is colored in magenta, and the gray stick denotes the rotation axis. The electrostatic surface of PHAb10 with two potential substrate-binding cavities are shown (negative = red; positive = blue). (**c**) Dimeric structure of PHAb11. Chain A is colored in wheat, Chain B is colored in green-cyan, and the red stick denotes the rotation axis. The electrostatic surface of PHAb11 with two potential substrate binding cavities is also shown

*Figure 4 continued on next page*

*Figure 4 continued*

(negative = red; positive = blue). (**d**) Superimposition of monomeric PHAb8, PHAb10, and PHAb11. PHAb8 is shown in light blue, PHAb10 in green, and PHAb11 in wheat. Residues involved in the catalytic triad are shown in sticks. (**e–f**) Residual bactericidal activity of PHAb10, PHAb11, and their truncation variants. Each truncation variant is treated at 100°C for 1 hr, stored at 25°C for 1 hr, and then tested for bactericidal activity by a log-killing assay. All assays were performed with at least three biological replicates (n=3-9). Dash lines represent the limit of detection and data below the limit of detection is not shown. (**g**) Thermal unfolding curves of PHAb8, PHAb10, and PHAb11 from 20°C to 100°C as determined by nano-differential scanning fluorimetry (nanoDSF). Values on the y-axis represent the first derivative of the fluorescence ratio at 350 nm and 330 nm. Peaks represent the transition temperature of each protein. (**h–j**) Circular dichroism (CD) spectra of PHAb8 (**h**), PHAb10 (**i**), and PHAb11 (**j**) at 90°C and temperatures close to their transition temperatures. (**k–o**) CD spectra of PHAb10, PHAb11, and their truncation variants before and after heat treatment. Each domain fragment is treated at 100°C for 1 hr, stored at 25°C for additional hour prior to CD detection.

The online version of this article includes the following figure supplement(s) for figure 4:

**Figure supplement 1.** Structural analysis of PHAb8, PHAb10, and PHAb11.

**Figure supplement 2.** Prophage analysis of PHAbs.

showed that for all strains tested, the holoenzyme and its antimicrobial peptide domain P10-CP retained high bactericidal activity, but its lysozyme catalytic domain P10-Lys did not (*Figure 4e*). Except for the N-terminal peptide P11-NP, other truncated PHAb11 still maintained substantial bactericidal activity after heating at 100°C for 1 hr (*Figure 4f*). These observations raised the question of whether PHAb10 and PHAb11 dimers are hyper-thermostable. To test this, we determined the temperature-dependent denaturation of these enzymes by nano-differential scanning fluorimetry (nanoDSF). Our results showed that PHAb8, PHAb10, and PHAb11 were denatured with transition temperatures of 41.4°C, 53.7°C, and 59.9°C, respectively (*Figure 4g*). Interestingly, signs of aggregation were only observed in PHAb8 but not in PHAb10 and PHAb11 (*Figure 4—figure supplement 1j*), indicating that PHAb10 and PHAb11 underwent structural refolding without precipitation during heat treatment. Consistent with these observations, the corresponding circular dichroism (CD) curves of PHAb8, PHAb10, and PHAb11 at 90°C were significantly different from their corresponding curves at temperature close to their transition temperature (*Figure 4h–j*), suggesting that, similar to PHAb8, PHAb10 and PHAb11 lost their native conformation at high temperature such as 90°C. To further understand the thermodynamics of PHAb10 and PHAb11, we examined the structure of PHAb10 before and after heat treatment at 100°C for 1 hr by CD. Intriguingly, only minor differences were observed before and after heat treatment (*Figure 4k*), indicating that PHAb10 almost refolded to its native conformation during cooling after heat treatment. The C-terminal antimicrobial peptide P10-CP of PHAb10 could withstand heat treatment, but P10-Lys was deformed after being treated at 100°C for 1 hr (*Figure 4l*). Similar to the folding-refolding phenomenon of PHAb10, PHAb11 and its N-terminal peptide-deleted variant P11-C could restore to their native conformation after 1 hr treatment at 100°C (*Figure 4m and n*), while its C-terminal peptide-deleted variant P11-N could not refold to its native conformation (*Figure 4o*), indicating that C-terminal peptide was essential for its hyper-thermostability. Altogether, these observations suggested that dimerized PHAb10 and PHAb11 possessed unique folding-refolding thermodynamic mechanism.

## Folding-refolding thermodynamic of PHAb10 dimer is governed by intermolecular interactions

Since PHAb10 dimer and PHAb11 dimer exhibit a similar folding-refolding thermodynamic profiles, we wondered whether it was achieved through dimer-monomer switching. Therefore, we used PHAb10 as an example to test this hypothesis. We carefully examined the topology of the PHAb10 dimer and found that it was supported by seven pairs of intermolecular H-bonds, with three pairs supporting the head, each two pairs in midbody and tail of the PHAb10 dimer (*Supplementary file 1d*). Specifically, in the top view of chain A, Arg20 and Gly27 were involved in intermolecular H-bond interactions, supporting the head region, Gly128 and Gly129-mediated interaction contributed to the stability of midbody, while Lys133 and Arg137 were involved in intermolecular H-bond forces anchoring the tails of the PHAb10 dimer (*Figure 5a*; *Supplementary file 1d*). Next, we constructed several PHAb10 variants involving depletion of intermolecular force at the head, midbody, tail, or entire entity of the PHAb10 dimer (*Supplementary file 1e*). Since PHAb10 was a verified dimer after chemical cross-linking in SDS-PAGE gel (*Figure 5b*), we further examined the polymerization of these PHAb10 variants by Native-PAGE. As shown in *Figure 5c*, wildtype PHAb10 showed two bands corresponding

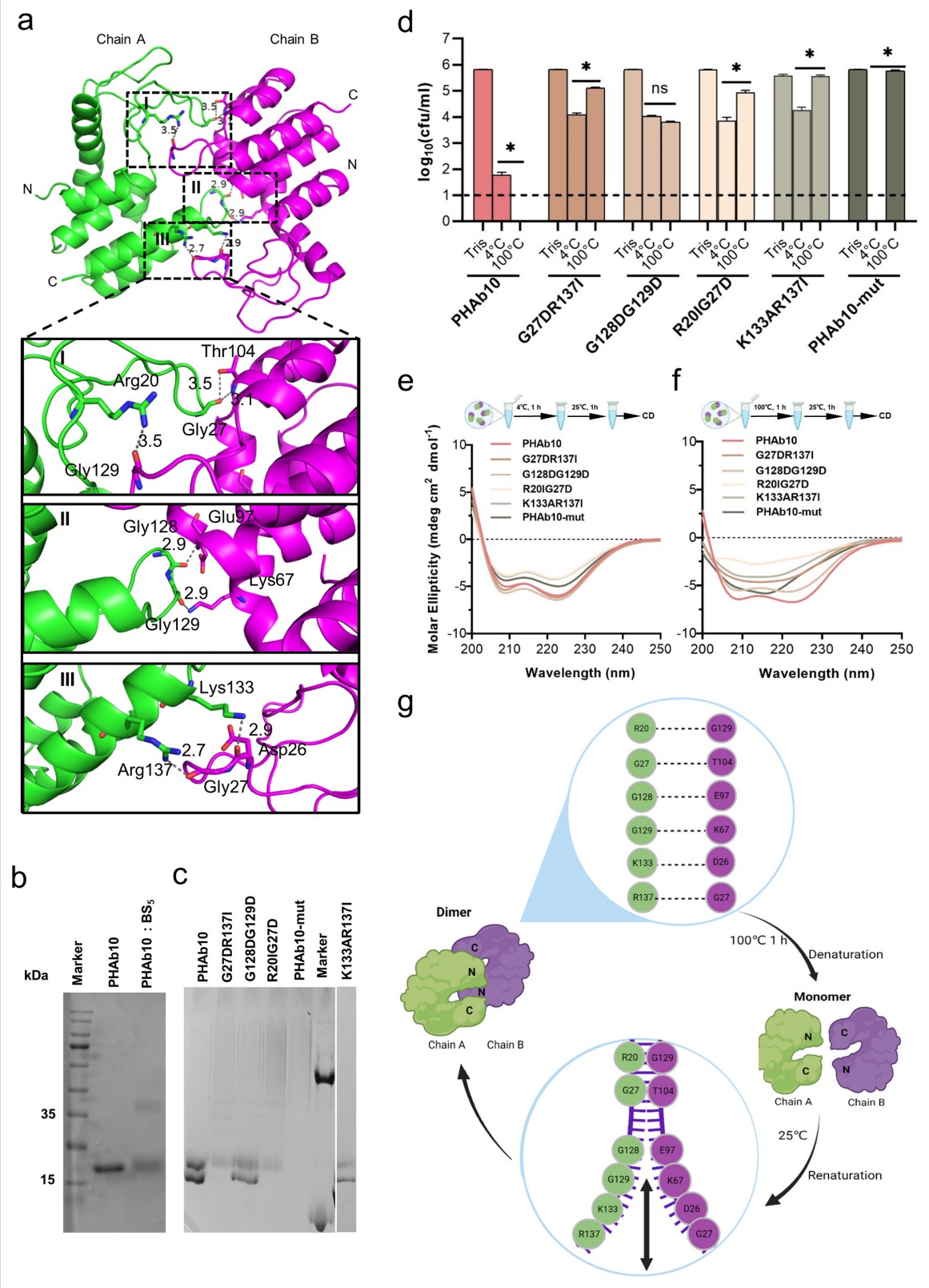

**Figure 5.** Folding-refolding thermodynamics of PHAb10. (**a**) Intermolecular interactions of PHAb10 dimer. The residues involved are shown as sticks. (**b**) SDS-PAGE analysis of PHAb10 with or without chemical cross-linker. (**c**) Native-PAGE analysis of PHAb10 and its variants. Marker: the upper band represents BSA (66.4 kDa), and the lower band represents lysozyme (14 kDa). (**d**) Bactericidal activity of PHAb10 and its variants before and after heat treatment. Each protein is incubated for 1 hr at 4°C or 100°C, stored at 25°C for an additional 1 hr, and then examined for bactericidal activity in 20 mM

*Figure 5 continued on next page*

*Figure 5 continued*

Tris-HCl (pH 7.4) at a concentration of 50 µg/ml against exponential *A. baumannii* 3437 cells for 1 hr. All assays were performed with at least three biological replicates (n=6-9). Dash lines represent the limit of detection and data below the limit of detection is not shown. Data are analyzed by two-tailed Student's t-tests. ns: statistically not significant; *: p<0.05. (**e–f**) Circular dichroism spectra of PHAb10 and its mutants before and after treatment at 100°C for 1 hr. (**g**) Zipper model showing the thermodynamics of PHAb10 dimer.

The online version of this article includes the following source data for figure 5:

**Source data 1.** PDF file containing original SDS/Native-PAGE gels for **Figure 5b and c**, indicating the relevant bands.

**Source data 2.** Original file for SDS/Native-PAGE analysis displayed in **Figure 5b and c**.

to its monomeric and dimeric forms. While the variants R20IG27D, which has mutations in the head-supporting site, and G27DR137I, which has mutations in both the head and tail supporting sites, lost dimerization ability. As a result, both variants lost most of their bactericidal activity after treatment at 100°C for 1 hr (**Figure 5d**), which could be further explained by the failure to recover their native forms after heat treatment (**Figure 5e and f**). In support of these observations, the variant PHAb10-mut containing mutations in all seven pairs of dimer-supporting sites showed aborted bactericidal activity (**Figure 5d**) and failed to recover its native conformation (**Figure 5e and f**) after treatment at 100°C for 1 hr. In contrast, the variants G128D/G129D with mutations in the midbody-supporting sites still showed a mixture of monomers and dimers (**Figure 5c**), indicating that depletion of the midbody-supporting force hardly affected the dimerization of PHAb10. Consistent with this observation, the variant G128D/G129D still remained intact bactericidal activity (**Figure 5d**) and could refold to its native form after heat treatment (**Figure 5e and f**). Moreover, the variant K133A/R137I with mutations in the tail-supporting site also showed aborted bactericidal activity (**Figure 5d**) and an irreversible conformation change after heat treatment (**Figure 5e and f**), although it maintained a weak dimerization ability (**Figure 5c**). This suggested that intermolecular forces at the tail of the PHAb10 dimer had only a small impact on its thermodynamics properties. Taken all these observations together, we proposed a model by which the PHAb10 dimer executed its folding-refolding thermodynamics (**Figure 5g**). At temperatures above its transition point (i.e. >53.7°C), dimerized PHAb10 dissociated into monomers without precipitation. During the cooling process, monomeric PHAb10 refolded to a stable dimer via the intermolecular interactions between the two monomers, i.e., three pairs of intermolecular bonds at the head, two in the midbody, and two at the tail of the PHAb10 dimer. The folding and refolding dynamics of PHAb10 after heat treatment were somewhat similar to the opening and closing of zippers, where the intermolecular bonds act as inducers.

## PHAb10 efficiently removes *A. baumannii* from infection sites in vivo

Since PHAb10 exhibited rare cytotoxicity at the cellular level (**Figure 6—figure supplement 1**), we further tested its bactericidal activity in two different mouse infection models. In a burn wound model of *A. baumannii* infection (**Figure 6a**), mice were first treated with 10 µg/mouse PHAb10 (10 µl; n=5) for 4 hr, and then infected with 10 µl 5×10⁸ cfu/ml of *A. baumannii* 3437 for 24 hr. We found significantly less viable bacteria remaining at the site of infection compared to Tris-treated controls (**Figure 6b**). Treatment with 4 µg/mouse minocycline (10 µl; n=5) also showed good bacterial removal compared to the control group, however, no statistical difference was observed between groups treated with 10 µg/mouse PHAb10 and 4 µg/mouse minocycline (**Figure 6b**), suggesting that when administered topically, PHAb10 was as efficient as minocycline in clearing susceptible bacteria from the site of infection. In a mouse abscess model, mice were infected hypodermically (i.h.) with 25 µl of *A. baumannii* 3437 at a concentration of 5×10⁸ cfu/ml for 24 hr, and then treated with 10 µg/mouse PHAb10 (10 µl; n=6) or 4 µg/mouse minocycline (10 µl; n=7) subcutaneously for 5 days. Six days after infection, viable bacteria were recovered from the infection site and counted (**Figure 6c**). Our results showed that there was rare difference between the minocycline-treated group and the Tris-treated control group (**Figure 6d**). In contrast, a significant reduction in residual *A. baumannii* was observed in the PHAb10-treated group (**Figure 6d**), indicating that PHAb10 had better bactericidal efficacy than minocycline when injected subcutaneously at the site of infection in a skin infection model. Notably, there is little difference in the change in animal body weight between groups treated with PHAb10, minocycline, or Tris buffer (**Figure 6—figure supplement 2**). Taking all these observations together,

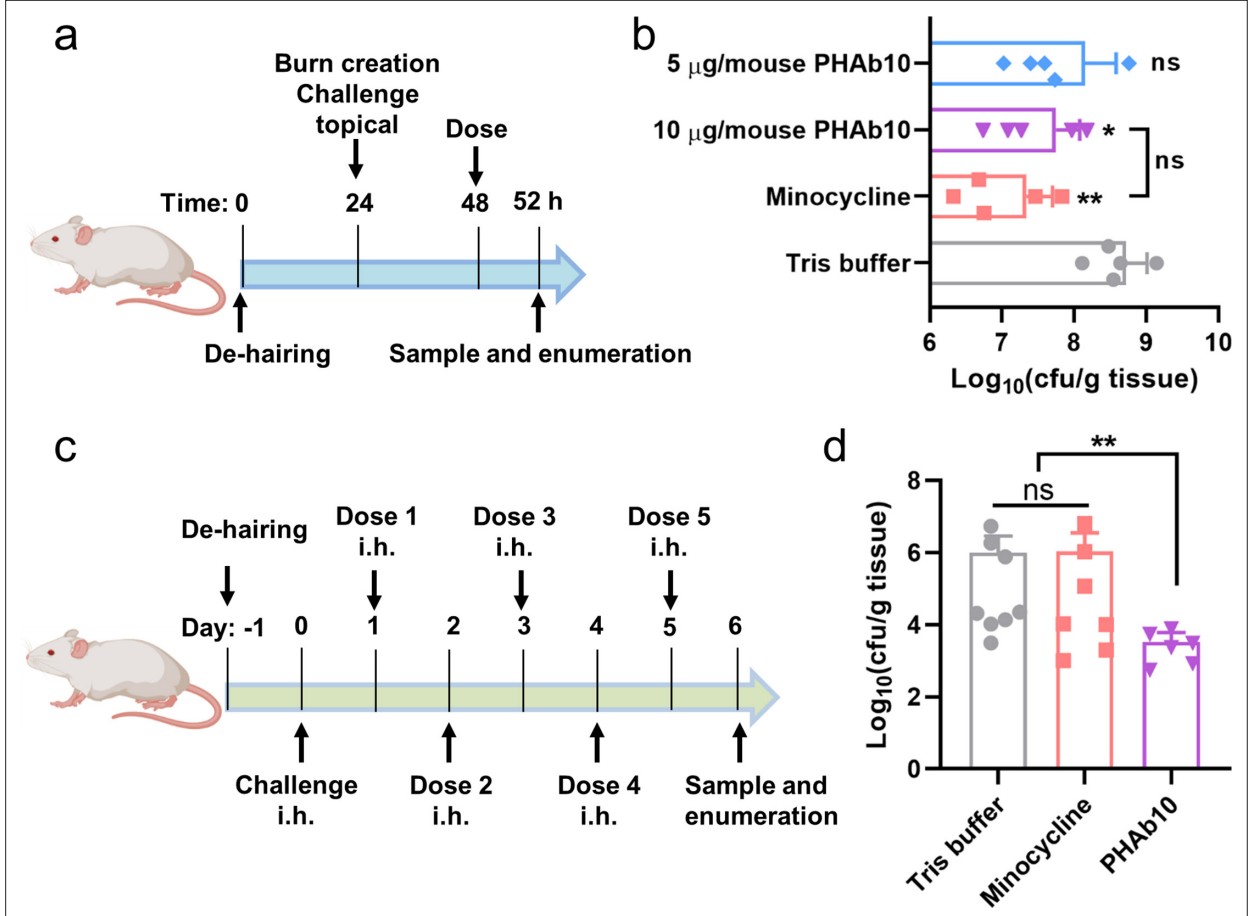

**Figure 6.** PHAb10 shows high efficacy in mouse infection models. (**a**) Experimental schema for burn wound infection model. Burn wounds produced by exposing naked skin to 65°C water for 12 s was infected with 10 µl 5×10⁸ cfu/ml of *A. baumannii* 3437. At 24 hr post-colonization, mice are treated with 5 or 10 µg/mouse PHAb10 (10 µl; n=5), 4 µg/mouse minocycline (10 µl; n=5), or an equal volume of Tris buffer (10 µl; n=5). Four hours post-treatment, viable bacteria on the burn wound skin is collected and counted. (**b**) Viable *A. baumannii* collected from burn wound skin after each treatment. The number of viable cells in each group is normalized and compared with the number of viable cells treated with Tris buffer by one-way analysis of variance (ANOVA). ns: statistically not significant; *: p<0.05; **: p<0.01. (**c**) Experimental schema for abscess model. Mice are infected hypodermically with 25 µl 5×10⁸ cfu/ml of *A. baumannii* 3437 on the right side of the back dorsum. Twenty-four hours after infection, mice are injected hypodermically with 10 µg/ mouse PHAb10 (10 µl; n=6), 4 µg/mouse minocycline (10 µl; n=7), or an equal volume of Tris buffer (10 µl; n=8). Each group is injected subcutaneously once a day for 5 consecutive days. Viable bacteria were counted at the infection site 24 hr after the last dose. (**d**) Viable *A. baumannii* collected from the skin after each treatment. The number of viable cells from each group was normalized and compared with those of the Tris buffer-treated group by ANOVA. ns: statistically not significant; **: p<0.01.

The online version of this article includes the following figure supplement(s) for figure 6:

**Figure supplement 1.** Relative viability of HepG2 cells exposed to different concentrations of PHAb10.

**Figure supplement 2.** Changes in mouse body weight in a mouse abscess model.

PHAb10 appears to exhibit good bacterial clearance efficacy in vivo and may represent a potential antimicrobial agent for the treatment of drug-resistant bacteria.

## Discussion

The uncontrollable spread and emergence of antimicrobial resistance requires the discovery of new antimicrobial agents with different targets or mechanisms of action than traditional antibiotics. Bacteriophage-derived peptidoglycan hydrolases (i.e. lysins) have been investigated as promising alternatives to combat resistant bacteria due to their unique mechanism of action, potent bactericidal activity, low risk of drug resistance development. However, for Gram-negative bacteria, the effects of phage-derived lysins were often hampered by their outer membranes, which requires

more strategies to overcome this additional barrier. Although some engineered lysins (especially Artilysins, Innolysins, and Lysocins) have shown promising performance, clinically promising lysins capable of destroying Gram-negative bacteria remain inadequately addressed. Considering the fact that lysins from Gram-negative bacteria usually contain antimicrobial peptide-like regions (*Vázquez et al., 2021*), in the present work, we report a new strategy to mine active bactericidal peptidoglycan hydrolases from bacterial proteomes through lysin-derived antimicrobial peptide-priming screening. As a proof-of-concept, we successfully discovered the two peptidoglycan hydrolases PHAb10 and PHAb11 from the proteome of *A. baumannii* using the PlyF307 lysin-derived peptide as a template, highlighting the utility of this strategy in mining antimicrobial agents from bacterial database.

Notably, PHAb10 and PHAb11 exhibited high bactericidal activity against a variety of Gram-negative and Gram-positive microbes. Unlike other Gram-negative lysins that act in an outer membrane permeabilizer-dependent manner (*Briers et al., 2011*; *Walmagh et al., 2013*; *Oliveira et al., 2014*; *Plotka et al., 2019*), PHAb10 and PHAb11 exhibited intrinsic activity against Gram-negative and Gram-positive pathogens via different mechanisms of action. Due to their C-terminal antimicrobial peptides P10-CP and P11-CP, PHAb10 and PHAb11 can penetrate the outer membrane of Gram-negative bacteria in a self-primed manner, allowing them to enter and cleave peptidoglycan bonds, ultimately leading to osmotic lysis of Gram-negative pathogens, while Gram-positive bacteria are killed mainly through the action of the intrinsic antimicrobial peptides.

Structure-based homology analysis revealed that PHAb10 and PHAb11 probably belonged to the T4 lysozyme family and possessed the conserved Glu-Asp-Thr catalytic triad shared by traditional phage lysins and bacterial autolysins, namely Glu17, Asp26, and Thr32 of PHAb10, and Glu52, Asp61, and Thr67 of PHAb11. It is known that T4 lysozyme (T4L) from T4 phage completely loses its enzymatical activity after treatment at 75°C for 5 min (*Tsugita and Inouye, 1968*), but somehow, heat-denatured T4L retains its bactericidal activity because of the amphipathic α-helix region at the C-terminus of the enzyme which contains membrane disturbing activity (*Düring et al., 1999*). Unlike T4L, whose monomer crystals have been exhaustively studied, PHAb10 or PHAb11 is a homodimer that retain almost 100% of their bactericidal activity after heat treatment through a unique dimer-monomer transition. Interestingly, several T4L mutants have been reported to be homodimers crystallographically (*Banatao et al., 2006*), although naturally occurring dimeric T4L-like enzymes are still rarely reported. Notably, heat treatment converts HEWL to a partially unfolded, enzymatically inactive and more hydrophobic dimeric form, which exhibits an enhanced bactericidal activity against Gram-negative bacteria (*Ibrahim et al., 1996*). The excellent thermostability capacity of PHAb10 and PHAb11 distinguishes them from most other enzymes reported to date. For example, gp36C lysin from bacteriophage φKMV, a known highly thermostable lysozyme, retains only ~50% residual activity after treatment at 100°C for 1 hr (*Lavigne et al., 2004*; *Briers et al., 2006*). Heat-resistant lysin has potential applications in animal feed premixes (pelletizing temperature 75–98°C) and medical devices requiring autoclaving.

To the best of our knowledge, few proteins have been demonstrated to have reversible folding-refolding thermodynamics at high temperature, but their mechanisms are unclear, such as HPL118, HPL511, and HPLP35 (*Schmelcher et al., 2012*). Here, we found that PHAb10 had unique temperature-dependent folding-refolding thermodynamic, and structural and biochemical analyses further suggested that PHAb10 undergoes a unique dimer-monomer transition mediated by seven pairs of intermolecular interactions to survive thermal treatment.

Bacteriophages are known to be the most abundant biological entities on earth (*Knowles et al., 2016*). Over billions of years of coevolution, more than 80% of bacterial genomes have become populated with at least one prophage (*Keen and Dantas, 2018*; *Miller-Ensminger et al., 2018*), also known as molecular imprints, and which can serve as an untapped arsenal for the discovery of phage-related antimicrobial agents. Notably, prophage identification lags greatly behind genome or metagenome sequencing. Therefore, there is reason to believe that advances in bacterial culturomics (*Lagier et al., 2018*), 'rebooting' phage genomes in more tractable hosts (*Kilcher et al., 2018*), and computational biology (*Arndt et al., 2019*; *Song et al., 2019*), may help discover otherwise inaccessible prophages from bacterial genome big data. In turn, this will increase the added value of bacterial genomes as a new source of unearthed therapeutically promising antimicrobial drugs. More importantly, our findings implied that big data, such as bacterial genomes and proteomes, human proteomes, and other

metagenomic datasets, could in principle become an encrypted power in the fight against antimicrobial resistance.

## Materials and methods

### Bacterial strains

The bacterial strains used in this work are described in *Supplementary file 1f*. *A. baumannii*, *E. coli*, *P. aeruginosa*, *K. pneumoniae*, *E. faecalis,* and *S. aureus* were all grown in Lysogeny Broth (LB) at 37°C. *S. pneumoniae* was statically cultured in Todd Hewitt Broth with 0.5% Yeast extract (THY) at 37°C with 5% $CO_2$. All other *Streptococci* were grown in Brain Heart Infusion broth at 37°C.

### Screening of putative peptidoglycan hydrolases from *A. baumannii*

Putative peptidoglycan hydrolases from the *A. baumannii* (PHAbs) proteome database were obtained by BLASTP in NCBI using the lysin PlyF307-derived antimicrobial peptide P307 as input sequence. Sequences marked as partial, shorter than 110 amino acids, longer than 300 amino acids, or clearly not belonging to lysins were manually excluded from further analysis. Multiple sequence alignments were performed by the ClustalW algorithm in MEGAX software and visualized by Jalview. Phylogenetic trees were constructed by MEGAX software using the neighbor-joining method and visualized by the iTOL online service (http://itol.embl.de/). Three-dimensional (3D) protein models showing different amino acid residues were predicted online by SWISS-MODEL (https://swissmodel.expasy.org/).

### In silico analysis of PHAbs

The basic physical and chemical properties of PHAbs were analyzed by ProtParam (https://web.expasy.org/protparam/). Solubility was predicted by Protein-Sol (https://protein-sol.manchester.ac.uk/), and signal peptides were predicted by SignalP (https://services.healthtech.dtu.dk/service.php?SignalP-5.0). The transmembrane area was predicted by TMHMM (https://services.healthtech.dtu.dk/service.php?TMHMM-2.0), while domain predictions were performed through CD searches in the NCBI database (https://www.ncbi.nlm.nih.gov/Structure/cdd/wrpsb.cgi) and InterPro in the EBI database (http://www.ebi.ac.uk/interpro/search/sequence/). The Percent Identity Matrix was calculated by Clustal Omega (https://www.ebi.ac.uk/Tools/msa/clustalo/) and visualized with TBtools. The prophage origination analysis was predicted by PhageBoost (https://phageboost.ku.dk/) (*Sirén et al., 2021*). The net charge of the N-terminal and C-terminal 30 amino acids of each PHAb was calculated by BaAMPs (http://www.baamps.it/tools/calculator) under neutral condition. The Local Net Charge of each PHAb was obtained by assigning 1 to the basic amino acids Arg (R) and Lys (K) and –1 to the acidic amino acids Glu (E) and Asp (D) and summing all assignments in a 31 amino acid window composing of 15 amino acids before and after the determined position. The 3D models of each PHAb were predicted by AlphaFold2 and visualized by PyMOL.

### Gene synthesis and cloning

The coding sequences of the original five putative peptidoglycan hydrolases, namely, PHAb7, PHAb8, PHAb9, PHAb10, and PHAb11, as well as the PHAb10 variant, PHAb10-mut, with six site mutations were chemically synthesized with codon optimization and cloned into a pET28b(+) vector by *Nco*I and *Xho*I restriction sites. Additional variants of PHAb10 and truncations of PHAb10 and PHAb11 were constructed by primer-derived overlap PCR (*Supplementary file 1g*). All constructs were electro-transformed into *E. coli* BL21(DE3) competent cells for expression. Peptides derived from PHAb10 and PHAb11, namely P10-CP (amino acids 111–149), P11-NP (amino acids 1–36), and P11-CP (amino acids 146–184), were chemically synthesized and dissolved in ultrapure water before use.

### Protein purification

BL21(DE3) cells were grown in LB to an optical density at 600 nm ($OD_{600}$) of 0.4–0.6 and induced with 0.2 mM isopropyl β-D-thiogalactopyranoside at 37°C for 2 hr. Cells were then harvested and lysed by sonication. Supernatant was collected and passed through a Ni-NTA column pre-equilibrated with 20 mM imidazole. Fractions were collected by washing and eluting with 60 mM and 250 mM imidazole, respectively. The collected active protein fractions were pooled, dialyzed against 20 mM Tris-HCl (pH 7.4), and then filter-sterilized. Protein for crystallization was further purified by size-exclusion

chromatography on a Superdex 75 (GE Healthcare) with gel filtration buffer (50 mM Tris-HCl, pH 8.0, 150 mM NaCl, 1 mM DTT). Protein purity was examined by 12% SDS-PAGE, and protein concentration determined using the BCA Protein Concentration Assay Kit. Dimerization of PHAb10 and its variants was detected by Native-PAGE.

## Halo assay

Bacterial strains were grown overnight in 100 ml LB, harvested, and resuspended in 50 ml phosphate buffered saline (PBS) containing 0.7% agarose. Cells were then autoclaved and poured into glass Petri dish culture plates. Ten microliter of peptidoglycan hydrolases at different concentrations (0, 2.5, 5, and 10 µg/ml) were dropped onto each bacterial lawn and incubated at 37°C till a clear zone was formed, and its area was determined using the method previously described (*Zhang et al., 2017*; *Vander Elst et al., 2020*). Wells treated with an equal volume of 20 mM Tris-HCl were used as controls.

## Antibacterial activity assay

To determine the antibacterial activity of peptidoglycan hydrolases, bacteria were cultured overnight (stationary phase, $OD_{600}=1.2–1.6$), then transferred to fresh medium at 100-fold dilution and continued to grow for 3 hr to $OD_{600}=0.5–0.6$ in the exponential growth phase. Cultures of different bacteria were centrifuged at 10,000×$g$ for 1 min, washed once and resuspended in 20 mM Tris-HCl, pH 7.4. Stationary phase bacterial cells were diluted to a final $OD_{600}$ of 0.6 before use. To test the bactericidal activity of each peptidoglycan hydrolase, 100 µl of a bacterial suspension in stationary phase or exponential phase was mixed with an equal volume of peptidoglycan hydrolase at different concentrations at 37°C for 1 hr with shaking at 200 rpm. The number of viable bacteria after each treatment was counted by plating serial dilutions on LB agar. Wells treated with an equal volume of dialysis buffer instead of peptidoglycan hydrolases were used as controls. Specifically, to test the dose-dependent antibacterial activity of PHAb10 and PHAb11, exponential cultures of various Gram-negative and Gram-positive bacteria were treated with 0, 0.625, 1.25. 2.5, 5, 6.25, 10, 12.5, 25, 50, or 100 µg/ml of each enzyme in 20 mM Tris-HCl (pH 7.4) at 37°C for 1 hr.

To test the time-killing curves of PHAb10 and PHAb11 against exponential *A. baumannii* 3437, bacterial suspensions were incubated with 50 µg/ml of each peptidoglycan hydrolase in 20 mM Tris-HCl (pH 7.4) for 0, 0.5, 1, 3, 5, 10, 15, 30,45, 60, and 120 min at 37°C.

To measure the effects of different environmental factors on the antibacterial activity of PHAb10 and PHAb11, exponential *A. baumannii* 3437 cells were treated with 50 µg/ml of each peptidoglycan hydrolase at different pH values (5, 6, 7, 8, 9, and 10), various temperatures (4°C, 25°C, 37°C, 45°C, 55°C, 65°C, 75°C, 85°C, and 100°C), and various NaCl concentrations (0, 25, 50, 100, 200, 500, and 1000 mM), and urea (0, 50, 100, 200, 500, and 1000 mM) at 37°C for 1 hr. All assays were performed at least three times in biological replicates.

## Thermostability assay

The thermostability of PHAb and their variants at 100 µg/ml was determined by a Prometheus NT.48 nanoDSF (NanoTemper Technologies, CA, USA) in the temperature range of 20–100°C (increasing step of 1 °C/min) in 20 mM Tris-HCl (pH 7.4). The first derivative of the fluorescence ratio at 350 and 330 nm (1st der, F350/F330) was calculated automatically by the PR-ThermControl software supplied with the instrumentation. The transition temperature (Tm) corresponds to the first-order peak of F350/F330.

## Circular dichroism

The CD spectra of PHAb and its variants at 200 µg/ml in 20 mM Tris-HCl (pH 7.4) were collected by an Applied Photophysics Chirascan Plus CD spectrometer (Leatherhead, UK) from 200 to 260 nm (0.1 cm path length) at room temperature. The spectra of air and Tris-HCl buffer (pH 7.4) were recorded as background and baseline, respectively. Secondary structures were calculated by CDNN V2.1 software supplied by the instrument manufacturer.

## Crystallization, data collection, and processing

In this study, crystals were obtained for each protein at a concentration of 20–25 mg/ml using sitting drop method at a 1:1 ratio on a reservoir of 0.1 M citric acid pH 3.5, 25% wt/vol polyethylene glycol

3350 after 1 week of incubation at 16°C. They were then immersed in a cryoprotectant consisting of a reservoir solution supplemented with 10% glycerol before being flash-frozen in liquid nitrogen. Diffraction data were collected at beamline 19U1 (BL19U1) at Shanghai Synchrotron Radiation Facility (SSRF), and the X-ray datasets were processed using HKL2000 and XDS software programs. The initial phase information was determined by molecular replacement in the program PHASER using *A. baumannii* AB 5075UW prophage (PDB code: 6ET6) as the initial search model. The structure was then improved by multiple rounds of manual construction and refinement by COOT and PHENIX. The final structures of PHAb8, PHAb10, and PHAb11 were visualized using PyMOL. Summary of data collection and refinement statistics are presented in *Supplementary file 1b*.

## Cell lines

HepG2 cells were provided by National Virus Resource Center, China (CSTR:16533.09.IVCAS9.092) and they were authenticated by STR profiling (ATCC Cell Line Authentication Service) and tested as mycoplasma negative by PCR (EZ-detect Mycoplasma Detection Kit). They were maintained in Dulbecco's modified Eagle medium (DMEM; Sigma-Aldrich, Shanghai, China) supplemented with 10% fetal bovine serum, 1% penicillin, and 1% streptomycin and cultured at 37°C and 5% $CO_2$.

## Cytotoxicity testing

The cytotoxicity of PHAb10 against HepG2 cells was determined by a Cell Counting Kit-8 (CCK-8) assay (Dojindo Molecular Technologies, Kumamoto, Japan) according to the manufacturer's protocol. HepG2 cells were seeded in 96-well plates at a density of 5000 cells per well in DMEM supplemented with 10% fetal bovine serum, 1% penicillin, and 1% streptomycin for 24 hr. The cells were then exposed to a series of concentrations of PHAb10 (0, 6.25, 12.5, 25, 50, and 100 µg/ml) for another 24 hr. Afterward, the contents of the plates were replaced with fresh medium containing 10% CCK-8 solution and incubated at 37°C for 1.5 hr. The final optical density at OD450 was noted by use of a microplate reader (SynergyH1; BioTek, USA). The results were expressed as relative cell viability, expressed as a percentage of the growth of cells in control wells treated with PBS only.

## Mouse experiments

All mouse infection experiments were conducted in an ABSL-2 lab, and all experimental methods were carried out in accordance with the regulations and guidelines set forth by the Animal Experiments Committee of the Wuhan Institute of Virology, Chinese Academy of Sciences. All experimental protocols were approved by the Animal Experiments Committee of Wuhan Institute of Virology, Chinese Academy of Sciences (WIVA17202102). During the experiment, animals were housed in individually ventilated cages following a range of animal welfare and ethical criteria and were euthanized at the end of observation. In a burn infection model, 6- to 8-week-old female BALB/c mice were anesthetized, shaved, and depilated to create a 2 cm² hairless area on the dorsum as previously described (*Li et al., 2021*). Partial thickness burn was achieved by exposing bare skin to water at 65°C for 12 s. The burns were then inoculated with 10 µl 5×10⁸ cfu/ml of *A. baumannii* 3437. 24 hr after colonization, mice were randomly separated into four groups and treated with 5 or 10 µg/mouse PHAb10 (10 µl; n=5), 4 µg/mouse minocycline (10 µl; n=5), or an equal volume of Tris buffer (10 µl; n=5). Four hours after treatment, mice were sacrificed by cervical dislocation to obtain skin. Specifically, the infected skin was excised, infiltrated in 1 ml PBST (PBS containing 0.1% Triton X-100) for 2 min, and homogenized using a NewZongKe MD1000 Tissue Cell-Destroyer (NewZongKe, Wuhan, Hubei, China). The resulting solution was serially diluted and plated on LB agar containing 4 µg/ml gentamicin and 2 µg/ml meropenem.

In a mouse abscess model, 6- to 8-week-old female mice were depilated to create a 2 cm² hairless area on the dorsum as described above and i.h. with 25 µl of *A. baumannii* 3437 at a concentration of 5×10⁸ cfu/ml on the right side of the back dorsum. After 24 hr of infection, mice were randomly divided into three groups and hypodermically treated with 10 µg/mouse PHAb10 (10 µl; n=6), 4 µg/mouse minocycline (10 µl; n=7), or an equal volume of Tris buffer (10 µl; n=8). Subcutaneous dosage was performed once a day for 5 consecutive days for each group. The body weight of mice in each group was monitored daily. Twenty-four hours after the last dosage, mice were sacrificed by cervical dislocation. Infected skin was excised, infiltrated in 1 ml PBST for 2 min, and subjected to tissue homogenization using a NewZongKe MD1000 Tissue Cell-Destroyer. Number of viable bacteria in

each group was enumerated by plating serial dilutions on LB agar containing 4 µg/ml gentamicin and 2 µg/ml meropenem.

## Statistical analysis

Two-tailed Student's t-tests were used to analyze all in vitro assays. In mouse models, viable bacterial cell numbers in each group were compared and analyzed by one-way analysis of variance (ANOVA). ns: statistically not significant; *: $p<0.05$; **: $p<0.01$.

## Acknowledgements

This work was supported by the National Natural Science Foundation of China (No. 32070187, 32161133003, 31770192, and 81802001), the Open Research Fund Program of National Bio-Safety Laboratory, Wuhan (No. 2021SPCAS001), and Wuhan Science and Technology Major Project (No. 2023020302020708). We thank the staff of BL19U1 beamline at National Center for Protein Sciences Shanghai and Shanghai Synchrotron Radiation Facility (Shanghai, China) for assistance during data collection. We thank Dr. Xuefang An and Dr. Yanfeng Yao from the Core Facility and Technical Support, Wuhan Institute of Virology for their assistance in animal experiments.

## Additional information

### Funding

| Funder | Grant reference number | Author |
|---|---|---|
| National Natural Science Foundation of China | 32070187 | Hang Yang |
| National Natural Science Foundation of China | 32161133003 | Hang Yang |
| National Natural Science Foundation of China | 31770192 | Hang Yang |
| National Natural Science Foundation of China | 81802001 | Fen Hu |
| Open Research Fund Program of National Bio-Safety Laboratory, Wuhan | 2021SPCAS001 | Hang Yang |
| Wuhan Science and Technology Major Project | 2023020302020708 | Jin He |

The funders had no role in study design, data collection and interpretation, or the decision to submit the work for publication.

### Author contributions

Li Zhang, Zirong Zhao, Data curation, Formal analysis, Investigation; Fen Hu, Data curation, Software, Formal analysis, Funding acquisition, Validation, Investigation, Visualization, Methodology; Xinfeng Li, Formal analysis, Project administration; Mingyue Zhong, Resources, Formal analysis, Project administration; Jiajun He, Software, Methodology; Fangfang Yao, Data curation, Investigation; Xiaomei Zhang, Formal analysis, Investigation; Yuxuan Mao, Formal analysis; Hongping Wei, Resources, Project administration; Jin He, Resources, Formal analysis, Supervision, Project administration; Hang Yang, Conceptualization, Formal analysis, Supervision, Funding acquisition, Methodology, Writing – original draft, Project administration, Writing – review and editing

### Author ORCIDs

Jin He (iD) https://orcid.org/0000-0002-1456-8284
Hang Yang (iD) https://orcid.org/0000-0001-6750-1465

### Ethics

All mouse infection experiments were conducted in an ABSL-2 lab, and all experimental methods were carried out in accordance with the regulations and guidelines set forth by the Animal Experiments

Committee of the Wuhan Institute of Virology, Chinese Academy of Sciences. All experimental protocols were approved by the Animal Experiments Committee of Wuhan Institute of Virology, Chinese Academy of Sciences (WIVA17202102).

Reviewer #1 (Public review): https://doi.org/10.7554/eLife.98266.3.sa1
Reviewer #2 (Public review): https://doi.org/10.7554/eLife.98266.3.sa2
Author response https://doi.org/10.7554/eLife.98266.3.sa3

## Additional files

### Supplementary files
• Supplementary file 1. Construction and characterization of PHAbs and their variants. (a) Physicochemical properties of putative antimicrobial peptides in PHAb10 and PHAb11. (b) Data collection and refinement statistics of PHAb8, PHAb10, and PHAb11 structures. (c) T4L-like lysozymes with characterized structures. (d) Intermolecular interactions of PHAb10 dimer. (e) Design of PHAb10 variants. (f) Bacterial strains used in this work. (g) Primers used in this study.

• MDAR checklist

### Data availability
The atomic coordinate and structure factors of PHAb8, PHAb10, and PHAb11 have been deposited in the PDB database with accession numbers 9KBT, 9KBQ and 9KBS, respectively.

The following datasets were generated:

| Author(s) | Year | Dataset title | Dataset URL | Database and Identifier |
|---|---|---|---|---|
| Hu F | 2024 | Crystal structure of PHAB8, a peptidoglycan hydrolase with relatively weak thermal stability and broad-spectrum | https://www.rcsb.org/structure/9KBT | RCSB Protein Data Bank, 9KBT |
| Hu F | 2024 | Crystal structure of PHAb10, a peptidoglycan hydrolase with thermal stability and broad-spectrum | https://www.rcsb.org/structure/9KBQ | RCSB Protein Data Bank, 9KBQ |
| Hu F | 2024 | Crystal structure of PHAb11, another peptidoglycan hydrolase with thermal stability and broad-spectrum | https://www.rcsb.org/structure/9KBS | RCSB Protein Data Bank, 9KBS |

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
