## [Editor Report · eLife assessment]

This **valuable** study explores a new strategy of lysin-derived antimicrobial peptide-primed screening to find peptidoglycan hydrolases from bacterial proteomes. Using this strategy, the authors identified five peptidoglycan hydrolases from Acinetobacter baumannii, which they tested on various Gram-positive and Gram-negative pathogens for antimicrobial activity. The revised manuscript addressed most of the prior concerns, and the data presented are **solid** and will be of interest to microbiologists.

---

## [Referee Report · Reviewer #1 (Public review)]

Summary:

Li Zhang et al. characterized two new Gram-negative endolysins identified through an AMP-targeted search in bacterial proteomes. These endolysins exhibit broad lytic activity against both Gram-negative and Gram-positive bacteria and retain significant antimicrobial activity even after prolonged exposure to high temperatures (100{degree sign}C for 1 hour). This stability is attributed to a temperature-reversible transition from a dimer to a monomer. The authors suggest several potential applications, such as complementing heat sterilization processes or being used in animal feed premixes that undergo high-temperature pelleting, which I agree with.

Strengths:

The claims are well-supported by relevant and complementary assays, as well as extensive bioinformatic analyses.

Weaknesses:

My last comments are minor and nearly all aim to improve the use of English language in the manuscript. However, a section describing the statistical analysis is still lacking. I believe that the presented manuscript can benefit from language editing, but I leave this decision with the editor.

---

## [Referee Report · Reviewer #2 (Public review)]

Summary:

The study explores a new strategy of lysin-derived antimicrobial peptide-primed screening to find peptidoglycan hydrolases from bacterial proteomes. Using this strategy authors identified five peptidoglycan hydrolases from A. baumannii. They further tested their antimicrobial activities on various Gram positive and Gram-negative pathogens.

Strengths:

Overall, the study is good and adds new members to the peptidoglycan hydrolases family. The authors also show that these lysins have bactericidal activities against both Gram-positive and Gram-negative bacteria. The crystal structure data is good, reveals different thermostablility to the peptidoglycan hydrolases. Structural data also reveals that PhAb10 and PHAb11 form thermostable dimer and data is corroborated by generating variant protein defective in supporting intermolecular bond pairs. The mice bacterial infection shows promise for the use of these hydrolases as antimicrobial agents.

Weaknesses:

While the authors have employed various mechanisms to justify their findings, some aspects are still unclear. Only CFU has been used to test bactericidal activity. This should also be corroborated by live/dead assay. Moreover, SEM or TEM analysis would reveal the effect of these peptidoglycan hydrolases on Gram-negative /Gram-positive cell envelopes. The authors claim that these hydrolases are similar to T4 lysozyme, but they have not correlated their findings with already published findings on T4 lysozyme. T4 lysozyme has C-terminal amphipathic helix with antimicrobial properties. Moreover, heat, denatured lysozyme also shows enhanced bactericidal activity due to the formation of hydrophobic dimeric forms, which are inserted in the membrane. Authors also observe that heat denatured PHAb10 and PHAb11 have bactericidal activity but no enzymatic activity. These findings should be corroborated by studying the effect of these holoenzymes/ truncated peptides on bacterial cell membranes. Also, a quantitative peptidoglycan cleavage assay should be performed in addition to halo assay. Including these details would make the work more comprehensive.

Revised version: The authors have addressed most of the questions in the revised version of the paper.

---

## [Author Response]

The following is the authors’ response to the original reviews.

**Public Reviews:**

**Reviewer #1 (Public Review):**
Summary:Li Zhang et al. characterized two new Gram-negative endolysins identified through an AMPtargeted search in bacterial proteomes. These endolysins exhibit broad lytic activity against both Gram-negative and Gram-positive bacteria and retain significant antimicrobial activity even after prolonged exposure to high temperatures (100{degree sign}C for 1 hour). This stability is attributed to a temperature-reversible transition from a dimer to a monomer. The authors suggest several potential applications, such as complementing heat sterilization processes or being used in animal feed premixes that undergo high-temperature pelleting, which I agree with.

We appreciate the reviewer’s valuable comments and suggestions.

Strengths:The claims are well-supported by relevant and complementary assays, as well as extensive bioinformatic analyses.

We appreciate the reviewer’s valuable comments and suggestions.

Weaknesses:There are numerous statements in the introduction and discussion sections that I currently do not agree with and consider need to be addressed. Therefore, I recommend major revisions.

Based on your valuable comments and suggestions, we have revised relevant introduction and discussion sections (pages 3-4, lines 82-101; page 21, lines 480-483).

Major comments:Introduction and Discussion:The introduction and the discussion are currently too general and not focused. Furthermore, there are some key concepts that are missing and are important for the reader to have an overview of the current state-of-the-art regarding endolysins that target gram-negatives. Specifically, the concepts of 'Artilysins', 'Innolysins', and 'Lysocins' are not introduced. Besides this, the authors do not mention other high-throughput mining or engineering strategies for endolysins, such as e.g. the VersaTile platform, which was initially developed by Hans Gerstmans et al. for one of the targeted pathogens in this manuscript (i.e., Acinetobacter baumannii). Recent works by Niels Vander Elst et al. have demonstrated that this VersaTile platform can be used to high-throughput screen and hit-to-lead select endolysins in the magnitude tens of thousands. Lastly, Roberto Vázquez et al. have recently demonstrated with bio-informatic analyses that approximately 30% of Gram-negative endolysin entries have AMP-like regions (hydrophobic short sequences), and that these entries are interesting candidates for further wet lab testing due to their outer membrane penetrating capacities. Therefore, I fully disagree with the statement being made in the introduction that endolysin strategies to target Gram-negatives are 'in its infancy' and I urge the authors to provide a new introduction that properly gives an overview of the Gram-negative endolysin field.

We thank the reviewer for the valuable suggestions. A new paragraph has been added to the revised manuscript to reflect the concepts and strategies for lysin engineering and discovery against Gram-negative bacteria (pages 3-4, lines 82-101).

Results:It should be mentioned that the halo assay is a qualitative assay for activity testing. I personally do not like that the size of the halos is used to discriminate in endolysin activity. In this reviewer's opinion, the size of the halo is highly dependent on (i) the molecular size of the endolysin as smaller proteins can diffuse further in the agar, and (ii) the affinity of the CBD subdomain of the endolysin for the bacterial peptidoglycan. It should also be said that in the halo assay, there is a long contact time between the endolysin and the bacteria that are statically embedded in the agar, which can result in false positive results. How did the authors mitigate this?

We quite agree with the reviewer that the halo assay is only a qualitative method for activity testing and may be perturbed by multiple parameters (DOI: 10.3390/antibiotics9090621). In our study, the halo assay was used only as a preliminary method to rapidly distinguish the activities of multiple candidates, and then the candidates with high antibacterial activities were further characterized through a series of in vitro and in vivo assays in this work.

Testing should have been done at equimolar concentrations. If the authors decided to e.g. test 50 µg/mL for each protein, how was this then compensated for differences in molecular weight? For example, if PHAb10 and PHAb11 have smaller molecular sizes than PHAb7, 8, and 9, there is more protein present in 50 µg/mL for the first two compared to the others, and this would explain the higher decrease in bacterial killing (and possibly the larger halos).

We thank the reviewer for his valuable suggestions and concerns. We agree with the reviewer that when we need to know exactly how much times more active an enzyme is than the another, we should directly compare the performance of the two enzymes at equimolar concentrations. In our previous work, we followed this rule to distinguish novel chimeric lysins from their parental lysins or their variants (DOI: 10.1128/AAC.00311-20; DOI: 10.1128/AAC.01610-19; DOI: 10.1128/AAC.02043-18). In the present work, our initial goal of testing was to reflect the robustness and efficiency of screening strategy initiated by lysinderived antimicrobial peptides. With this in mind, we therefore did not spend more effort to compare the activities of these candidates in detail but continued to clarify their host range and thermo-tolerance mechanisms, and then continued to examine their performance in infection models. Nonetheless, in future work, we will definitely follow your suggestions when it is necessary to quantify the differences between these candidates.

**Reviewer #2 (Public Review)**:Summary:The study explores a new strategy of lysin-derived antimicrobial peptide-primed screening to find peptidoglycan hydrolases from bacterial proteomes. Using this strategy authors identified five peptidoglycan hydrolases from A. baumannii. They further tested their antimicrobial activities on various Gram-positive and Gram-negative pathogens.

We appreciate the reviewer’s valuable comments.

Strengths:Overall, the study is good and adds new members to the peptidoglycan hydrolases family. The authors also show that these lysins have bactericidal activities against both Gram-positive and Gram-negative bacteria. The crystal structure data is good, and reveals different thermostablility to the peptidoglycan hydrolases. Structural data also reveals that PhAb10 and PHAb11 form thermostable dimers and data is corroborated by generating variant protein defective in supporting intermolecular bond pairs. The mice bacterial infection shows promise for the use of these hydrolases as antimicrobial agents.

We appreciate the reviewer’s valuable comments and suggestions.

Weaknesses:While the authors have employed various mechanisms to justify their findings, some aspects are still unclear. Only CFU has been used to test bactericidal activity. This should also be corroborated by live/dead assay. Moreover, SEM or TEM analysis would reveal the effect of these peptidoglycan hydrolases on Gram-negative /Gram-positive cell envelopes. The authors claim that these hydrolases are similar to T4 lysozyme, but they have not correlated their findings with already published findings on T4 lysozyme. T4 lysozyme has a C-terminal amphipathic helix with antimicrobial properties. Moreover, heat, denatured lysozyme also shows enhanced bactericidal activity due to the formation of hydrophobic dimeric forms, which are inserted in the membrane. Authors also observe that heat-denatured PHAb10 and PHAb11 have bactericidal activity but no enzymatic activity. These findings should be corroborated by studying the effect of these holoenzymes/ truncated peptides on bacterial cell membranes. Also, a quantitative peptidoglycan cleavage assay should be performed in addition to the halo assay. Including these details would make the work more comprehensive.

We thank the reviewer for his valuable suggestions and concerns. We agree with the reviewer that employing more methods and techniques such as SEM, TEM, live/dead imaging, and GC-MS will provide a deeper understanding of how these peptidoglycan hydrolases interact with the bacterial envelopes and peptidoglycan bones, which will definitely make our study more comprehensive. The principal idea of this study is, however, to test the robustness and effectiveness of the screening strategy triggered by lysin-derived antimicrobial peptide in discovering new peptidoglycan hydrolases. Therefore, we did not put more efforts in charactering the interactions of these peptidoglycan hydrolases with the bacterial envelopes/membranes in multiple assays; instead, we continued to elucidate their host range and thermo-tolerance mechanisms and then continued to examine their performance in infection models.

We are also very grateful to the reviewers for their suggestions to correlate our results to published findings on lysozymes. Based on these suggestions, we have included an extensive discussion in the Discussion section of the revised manuscript (page 22, lines 502-514).

**Recommendations for the authors:**

**Reviewer #1 (Recommendations For The Authors):**
Abstract and title.In my opinion, the current title does not fully cover the work that is presented in the manuscript.

According to your valuable comment, we have revised the title to “Dimer-monomer transition defines a hyper-thermostable peptidoglycan hydrolase mined from bacterial proteome by lysin-derived antimicrobial peptide-primed screening”.

Please remove the word 'novel' from the title, as well as elsewhere in the manuscript. As it is true that PHAb10 and PHAb11 are new, they are not novel. There are many reports that have been published on endolysins with activity against Gram-negatives, and sometimes even also Gram-positives.

We have changed the description of PHAb10 and PHAb11 to avoid using the word “novel”, but alternatively, using “new” or “active” in the title and throughout the text in the revised manuscript.

Additional information for the Introduction section in the Public Review:DOI: 10.1128/AAC.00285-16DOI: 10.1038/s41598-020-68983-3DOI: 10.1128/AAC.00342-19DOI: 10.1126/sciadv.aaz1136DOI: 10.1111/1751-7915.14339DOI: 10.1128/JVI.00321-21 G-

We appreciate the reviewer for these selected references and have cited almost all of them in a new paragraph in the Introduction section of the revised manuscript (pages 3-4, lines 82-101).

Minor Comments:Line 30. For a lay person it is not clear what is meant by 'unique mechanism of action.'

These has been replaced by “direct peptidoglycan degradation activity” in the revised manuscript (page 2, lines 30-31).

Line 60 & 62. Please merge these sentences into one as they have the same meaning.

We have deleted one of the sentences based on your suggestion.

Line 67. Replace 'also' with 'simultaneously'.

Revised as suggested (page 3, line 66).

Line 74. 'Modern clinical practice' should specifically refer to infectious diseases in humans.

Revised as suggested (page 3, line 73).

Line 76 to 105. There is too much information that is not focused. This section should be rewritten so that it is in line with the focus of the presented work. I would remove this section and replace it with a new section as proposed in my major comments.

Based on your suggestion, we deleted this section and prepared a new paragraph in the revised manuscript (pages 3-4, lines 82-101).

Line 113. I strongly disagree with the wording 'in its infancy'. Please see my major comment.

We have rewritten the paragraph as “However, compared with the current progress in the clinical translation of lysins against Gram-positive bacteria, the discovery of lysins against Gram-negative bacteria that meet the needs described in the WHO priority pathogen list is still urgently needed.” according to your valuable comments in the revised manuscript (page 4, lines 98-101).

Line 116. Remove 'on'.

Revised as suggested (page 4, line 104).

Results.Additional information for the Results section in the Public Review:DOI: 10.3390/antibiotics9090621

We thank the reviewer for this valuable reference, which has been cited in the Results section and Methods sections of the revised manuscript (page 7, line 159; page 25, line 605).

Minor comments:Line 135. Replace 'would' with 'could'.

Revised as suggested (page 5, line 124).

Line 150. Why was this naming decided to go from 11 -> 7, whereas in Figure 1a the clades go from I to V? This way of naming is not clear to me.

Thank you for the reviewer's question. There are two numbering systems here: 1-11 is the numbering of peptidoglycan hydrolases mined from different bacterial proteomes by lysin-derived peptide primer screening strategy, and the characterization of candidates mined from the proteome of A. baumannii are 7 to 11 (characterization candidates numbered 1 to 6 are from other bacterial proteomes). Whereas the cladistic analysis of all potential candidates in the A. baumannii proteome is regularly labelled by clade I to V.

Line 250. Replace 'casts doubt' with 'questions'.

Revised as suggested (page 10, line 244).

Line 252 to 257. I would encourage the authors to mention if there is any homology in between the peptides on the one hand, and in between the lysozyme catalytic domains on the other hand.

This information has been added to the revised manuscript (page 10, lines 249-251).

Line 266. The following sentence should be reworded: 'However, rare lytic activity was observed in P11-NP, suggesting that a potential role for it in functions other than bactericidal

activity.'

In the revised manuscript (page 11, lines 261-262), the sentence has been revised as “However, rare lytic activity was observed in P11-NP, suggesting that its function remains to be established”.

Line 276. Replace 'asked' with 'questioned'.

Revised as suggested (page 11, line 270).

From 302 onwards. Why was it chosen to solve the crystal structure of PHAb8, and not PHAb7 and 9? This should be briefly mentioned.

Initially we tried to decipher the structures of all five enzymes, but we finally obtained the crystal structures of only three enzymes, PHAb8, PHAb10, and PHAb11 by Xray crystallography. This reason has been added in the revised manuscript (page 13, lines 300301).

Line 437. Replace 'the burn wound model' with 'a burn wound model'.

Revised as suggested (page 19, line 433).

Line 445. Replace 'the mouse abscess model' with 'a mouse abscess model'.

Revised as suggested (page 19, line 441).

Line 449 to 451. Given that the mice received 5 doses of minocycline and no difference was observed with the group that received tris buffer, was it tested if the Acinetobacter baumannii 3437 isolates became resistant against minocycline during the experiment?

We appreciate the Reviewer for his valuable concern. In our study, we did not explore in detail the reasons why minocycline was ineffective. But we strongly agree with the reviewer that drug resistance may be one of the reasons.

Discussion.Minor comments.Line 479. Delete this sentence: 'Policy makers, scientists, enterprisers, and investigators have worked together for decades to exploit the 'trojan horse' globally, but new options for treating antimicrobial resistance in the clinic remain to be seen'.

Revised as suggested.

Line 483. Reformulate as follows: 'unique mechanism of action, potent bactericidal activity, low risks of drug resistance, and ongoing clinical trials targeting Gram-positive bacteria.' To my knowledge, all these clinical trials target *S. aureus*, but I might be wrong.

Revised as suggested (page 21, lines 476-478).

Line 486. 'However, for Gram-negative bacteria, the effects of phage-derived lysins were often hampered by their outer membranes, which requires more strategies to overcome this barrier.' After this sentence, the concepts of Artilysins, Innolysins, and Lysocins should be mentioned, in addition to the introduction. These are important engineering strategies and the reader should be informed that your strategy is thus not the only existent one.

Revised as suggested (page 21, lines 480-482).

Line 491. Please, again refer to the work of Roberto Vázquez et al., who has done very similar work to your work presented. DOI: 10.1128/JVI.00321-21

We have cited this interesting work in the Introduction section and Discussion section of the revised manuscript (page 4, line 106; page 21, lines 482-483).

Line 499. Reformulate: 'Gram-positive bacteria are primarily killed through the action of the antimicrobial peptides only'.

According to your suggestion, it was changed to “while Gram-positive bacteria are killed mainly through the action of the intrinsic antimicrobial peptides” in the revised manuscript (pages 21-22, lines 497-498).

Line 500. Delete this sentence, as this is already mentioned in the results and too detailed:'Interestingly, we noted a difference in the killing of Gram-positive bacteria by PHAb10 and PHAb11, which may be due to the fact that P11-CP had one more basic amino acid than P10CP, so it had stronger bactericidal activity.'

Revised as suggested.

Line 503. This statement doesn't make sense because you cannot directly compare ug/mL between endolysins, you must compare equimolar concentrations. Furthermore, testing conditions between studies were different, thus making this claim unjustified.

These statements have been deleted in the revised manuscript.

Line 524. Please delete:' To our knowledge, this is the first time that an enzyme had been found to adapt to ambient temperature by altering its dimerization state.'

Revised as suggested.

Figures.Figure 1a. Please choose a different name for 'dry job' and 'wet job'.

Following your suggestion, they have been specified as “In silico analysis” and “Experimental verification” in the revised Figure 1a.

Figure 6. I suggest moving Figure 6e to the supplementary materials and reorganizing Figure 6 with only panels a to d.

Revised as suggested.

Materials and Methods, References, and Supplementary Materials. No comments.Reviewer #2 (Recommendations For The Authors):Most figure labelings are very small and difficult to read.

All figures in the revised manuscript have been replaced with high-resolution figures, which hopefully will make these labels easier to follow.

The authors should include a data availability statement in the manuscript.

Revised as suggested (page 28, lines 704-706).